# PDGFRα signaling regulates Srsf3 transcript binding to affect PI3K signaling and endosomal trafficking

Thomas E Forman[1,2], Marcin P Sajek[3,4,5], Eric D Larson[6,7], Neelanjan Mukherjee[3,4], Katherine A Fantauzzo[1,4]*

[1]Department of Craniofacial Biology, School of Dental Medicine, University of Colorado Anschutz Medical Campus, Aurora, United States; [2]Medical Scientist Training Program, University of Colorado Anschutz Medical Campus, Aurora, United States; [3]Department of Biochemistry and Molecular Genetics, School of Medicine, University of Colorado Anschutz Medical Campus, Aurora, United States; [4]RNA Bioscience Initiative, University of Colorado Anschutz Medical Campus, Aurora, United States; [5]Institute of Human Genetics, Polish Academy of Sciences, Poznan, Poland; [6]Department of Otolaryngology – Head and Neck Surgery, University of Colorado Anschutz Medical Campus, Aurora, United States; [7]Basic and Translational Sciences, Penn Dental Medicine, University of Pennsylvania, Philadelphia, United States

*For correspondence:
katherine.fantauzzo@cuanschutz.edu

## eLife Assessment

This **fundamental** work provides new mechanistic insight in regulation of PDGF signaling through splicing controls. The evidence is **compelling** to demonstrate functional involvement of Srsf3, an RNA-binding protein, to this new and interesting mechanism. The work will be of broad interest to developmental biologists in general and molecular biologists/biochemists in the field of growth factor signaling and RNA processing.

**Abstract** Signaling through the platelet-derived growth factor receptor alpha (PDGFRα) plays a critical role in craniofacial development. Phosphatidylinositol 3-kinase (PI3K)/Akt is the primary effector of PDGFRα signaling during mouse skeletal development. We previously demonstrated that Akt phosphorylates the RNA-binding protein serine/arginine-rich splicing factor 3 (Srsf3) downstream of PI3K-mediated PDGFRα signaling in mouse embryonic palatal mesenchyme (MEPM) cells, leading to its nuclear translocation. We further showed that ablation of *Srsf3* in the murine neural crest lineage results in severe midline facial clefting and widespread alternative RNA splicing (AS) changes. Here, we demonstrated via enhanced UV-crosslinking and immunoprecipitation of MEPM cells that PDGF-AA stimulation leads to preferential binding of Srsf3 to exons and loss of binding to canonical Srsf3 CA-rich motifs. Through the analysis of complementary RNA-seq data, we showed that Srsf3 activity results in the preferential inclusion of exons with increased GC content and lower intron to exon length ratio. We found that Srsf3 activity downstream of PDGFRα signaling leads to retention of the receptor in early endosomes and increases in downstream PI3K-mediated Akt signaling. Taken together, our findings reveal that growth factor-mediated phosphorylation of an RNA-binding protein underlies gene expression regulation necessary for mammalian craniofacial development.

## Introduction

Craniofacial development is a complex morphogenetic process that requires a precise interplay of multiple cell and tissue types to generate the frontonasal skeleton. Disruption of this process can result in some of the most common birth differences in humans, such as cleft lip and palate (*Mai et al., 2019*). Signaling through the platelet-derived growth factor receptor alpha (PDGFRα) receptor tyrosine kinase (RTK) is essential for human craniofacial development. Heterozygous missense mutations in the coding region of *PDGFRA* that alter amino acids within the extracellular, transmembrane, or cytoplasmic domains of the receptor, in addition to single base-pair substitutions in the 3' untranslated region (3' UTR), are associated with nonsyndromic cleft palate (*Rattanasopha et al., 2012*). Further, single-nucleotide polymorphisms that repress transcriptional activity of the promoter upstream of *PDGFC*, which encodes one of two PDGFRα ligands, are associated with cleft lip and palate (*Choi et al., 2009*). This role of PDGFRα signaling in craniofacial development is conserved in mice, as *Pdgfra* mutant mouse models display a variety of defects that range from cleft palate to complete facial clefting (*Fantauzzo and Soriano, 2014*; *He and Soriano, 2013*; *Klinghoffer et al., 2002*; *Soriano, 1997*; *Tallquist and Soriano, 2003*). These phenotypes are recapitulated in embryos lacking both *Pdgfa* and *Pdgfc* (*Ding et al., 2004*). Phosphatidylinositol 3-kinase (PI3K) is the primary effector of PDGFRα signaling during skeletal development in the mouse (*Klinghoffer et al., 2002*). Following activation, PI3K increases phosphatidylinositol-3,4,5-trisphosphate (PIP$_3$) levels at the cell membrane, leading to the recruitment and subsequent phosphorylation of the serine/threonine kinase Akt. Akt subsequently dissociates from the membrane to phosphorylate an array of target proteins that are involved in wide-ranging cellular processes (*Manning and Cantley, 2007*). We previously identified proteins phosphorylated by Akt downstream of PI3K-mediated PDGFRα signaling in primary mouse embryonic palatal mesenchyme (MEPM) cells (*Fantauzzo and Soriano, 2014*). Gene ontology analysis revealed that 25% of the 56 proteins were involved in RNA processing, with a particular enrichment for RNA splicing (*Fantauzzo and Soriano, 2014*).

Alternative RNA splicing (AS) is a process by which different combinations of exons from the same gene are incorporated into mature RNA transcripts, thereby contributing to gene expression regulation and enhancing the diversity of protein isoforms (*Licatalosi and Darnell, 2010*). AS occurs in approximately 95% of multi-exon human genes, frequently in a tissue-specific manner (*Pan et al., 2008*; *Wang et al., 2008*). Dysregulation of AS causes a number of diseases due to mutations in precursor RNA sequences, mutations in core components of the spliceosome complex, and/or mutations in auxiliary RNA-binding proteins (RBPs) (*Scotti and Swanson, 2016*). These *trans*-acting auxiliary RBPs bind to specific sequence and/or structural motifs in a target RNA via one or more RNA-binding domains to promote or inhibit exon inclusion (*Fu and Ares, 2014*; *Licatalosi and Darnell, 2010*). The phenotypes resulting from global and/or tissue-specific knockout of multiple RBPs have established that RBP-mediated AS is an essential process during mouse craniofacial development (*Bebee et al., 2015*; *Cibi et al., 2019*; *Dennison et al., 2021*; *Forman et al., 2021*; *Lee et al., 2020*). We previously demonstrated that ablation of *Srsf3* in the murine neural crest lineage results in severe midline facial clefting and facial bone hypoplasia, due to defects in proliferation and survival of cranial neural crest cells, and widespread AS changes (*Dennison et al., 2021*).

Srsf3 belongs to the serine/arginine-rich (SR) protein family of splicing factors that generally promote exon inclusion by binding to exonic and intronic splicing enhancers and by recruiting spliceosome components to the 5' and 3' splice sites (*Fu and Ares, 2014*; *Licatalosi and Darnell, 2010*). Srsf3 specifically was shown to bind pyrimidine-rich motifs, with a preference for exonic regions (*Akerman et al., 2009*; *Änkö et al., 2012*). Srsf3 is phosphorylated downstream of PDGF and EGF stimulation and all-*trans* retinoic acid treatment, and by the kinases Akt and SRPK2 (*Bavelloni et al., 2014*; *Dennison et al., 2021*; *Fantauzzo and Soriano, 2014*; *Long et al., 2019*; *Zhou et al., 2012*). Phosphorylation of the Akt consensus sites within the C-terminal arginine/serine-rich (RS) domain of Srsf3 drives its translocation to the nucleus, where AS takes place (*Bavelloni et al., 2014*; *Dennison et al., 2021*; *Long et al., 2019*). Moreover, phosphorylation of the RS domain can alter the ability of SR proteins to interact with other proteins, such as the U1 small nuclear ribonucleoprotein (snRNP) component of the spliceosome and additional RBPs, and affect the ability of SR proteins to bind RNA (*Huang et al., 2004*; *Shen and Green, 2006*; *Shin et al., 2004*; *Xiao and Manley, 1997*). However, the molecular mechanisms by which Srsf3 activity is regulated downstream of specific signaling inputs in a context-specific manner to regulate RNA binding and/or sequence specificity remain undetermined.

Here, we identified changes in Srsf3-dependent AS and Srsf3 transcript binding in the absence or presence of PDGF-AA ligand in MEPM cells. RNA-sequencing (RNA-seq) analysis revealed that Srsf3 activity and PDGFRα signaling have more pronounced effects on AS than gene expression, as well as a significant dependence in regulating AS. Using enhanced UV-crosslinking and immunoprecipitation (eCLIP), we found a shift from intronic to exonic Srsf3 binding and loss of CA-rich sites upon PDGF-AA stimulation. Further, we demonstrated that the subset of transcripts that are bound by Srsf3 and undergo AS upon PDGFRα signaling commonly encode regulators of PI3K signaling and early endosomal trafficking, ultimately serving as a feedback mechanism to affect trafficking of the receptors. Combined, our findings provide significant insight into the mechanisms underlying RBP-mediated gene expression regulation in response to growth factor stimulation within the embryonic mesenchyme.

## Results

### PDGFRα signaling for 1 hr minimally affects gene expression

To determine the Srsf3-dependent changes in gene expression and AS downstream of PDGFRα signaling, we stably integrated a scramble shRNA (scramble) or an shRNA targeting the 3' UTR of *Srsf3* (shSrsf3) into immortalized MEPM (iMEPM) cells via lentiviral transduction (*Figure 1A*). Western blotting revealed a 66% decrease in Srsf3 protein levels in the shSrsf3 cell line (*Figure 1B*). We previously demonstrated that phosphorylated Srsf3 levels peaked in the nucleus of iMEPM cells following 1 hr of PDGF-AA ligand treatment (*Dennison et al., 2021*). As such, cells were left unstimulated (-PDGF-AA) or treated with 10 ng/mL PDGF-AA for 1 hr (+PDGF-AA). RNA was isolated and sequenced from three biological replicates across each of the four conditions (*Figure 1A*). We first examined Srsf3-dependent differentially expressed (DE) genes by comparing scramble versus shSrsf3 samples across the same PDGF-AA stimulation condition (-PDGF-AA or +PDGF-AA). We detected 827 DE genes in the absence of ligand treatment and 802 DE genes upon ligand stimulation (*Figure 1C*). There was extensive overlap (521 out of 1108; 47.0%) of Srsf3-dependent DE genes across ligand treatment conditions, resulting in a total of 1108 unique genes within both datasets (*Figure 1C and D*, *Figure 1—figure supplement 1A*). Of the 521 shared DE genes, 514 (98.7%) had the same directionality, including 273 (52%) with shared increases in expression in the shSrsf3 samples and 241 (46%) with shared decreases in expression in the shSrsf3 samples (*Figure 1C and D*). These findings demonstrate that expression of a set of genes (521) depends on Srsf3 activity independent of PDGFRα signaling, while a similarly sized set of genes (587) is differentially expressed in response to both Srsf3 activity and PDGFRα signaling (*Figure 1C and D*). Gene ontology (GO) analysis of the Srsf3-dependent DE genes revealed that the most significant terms for biological process commonly involved regulation of osteoblast differentiation, calcium-dependent cell–cell adhesion, regulation of cell migration, and canonical Wnt signaling, while only a handful of significant terms for molecular function were detected in unstimulated cells, relating to cation channel activity and phosphatase activity (*Figure 1—figure supplement 2A and B*).

We next examined PDGF-AA-dependent DE genes by comparing -PDGF-AA versus +PDGF-AA samples across the same Srsf3 condition (scramble or shSrsf3). We detected only 37 DE genes in scramble cells and 14 DE genes in shSrsf3 cells (*Figure 1E*). There was limited overlap (4 out of 47; 8.51%) of PDGF-AA-dependent DE genes across Srsf3 conditions, resulting in a total of 47 unique genes within both datasets (*Figure 1E and F*, *Figure 1—figure supplement 1B*). Of the four shared DE genes, three (75%) had shared increases in expression upon PDGF-AA stimulation (*Figure 1E and F*). These findings demonstrate that, unlike Srsf3 activity, PDGFRα signaling minimally affects gene expression at 1 hr of ligand stimulation, consistent with our previous findings in mouse embryonic facial mesenchyme (*Dennison et al., 2021*). Further, these results show that expression of a set of genes (4) depends on PDGFRα signaling independent of Srsf3 activity, while a larger set of genes (43) is differentially expressed in response to both PDGFRα signaling and Srsf3 activity (*Figure 1E and F*). GO analysis of the PDGF-AA-dependent DE genes revealed significant terms for biological process in the scramble cells relating to cell migration, response to growth factor stimulation, and regulation of transcription (*Figure 1—figure supplement 2A*). The most significant terms for molecular function commonly involved DNA binding (*Figure 1—figure supplement 2B*).

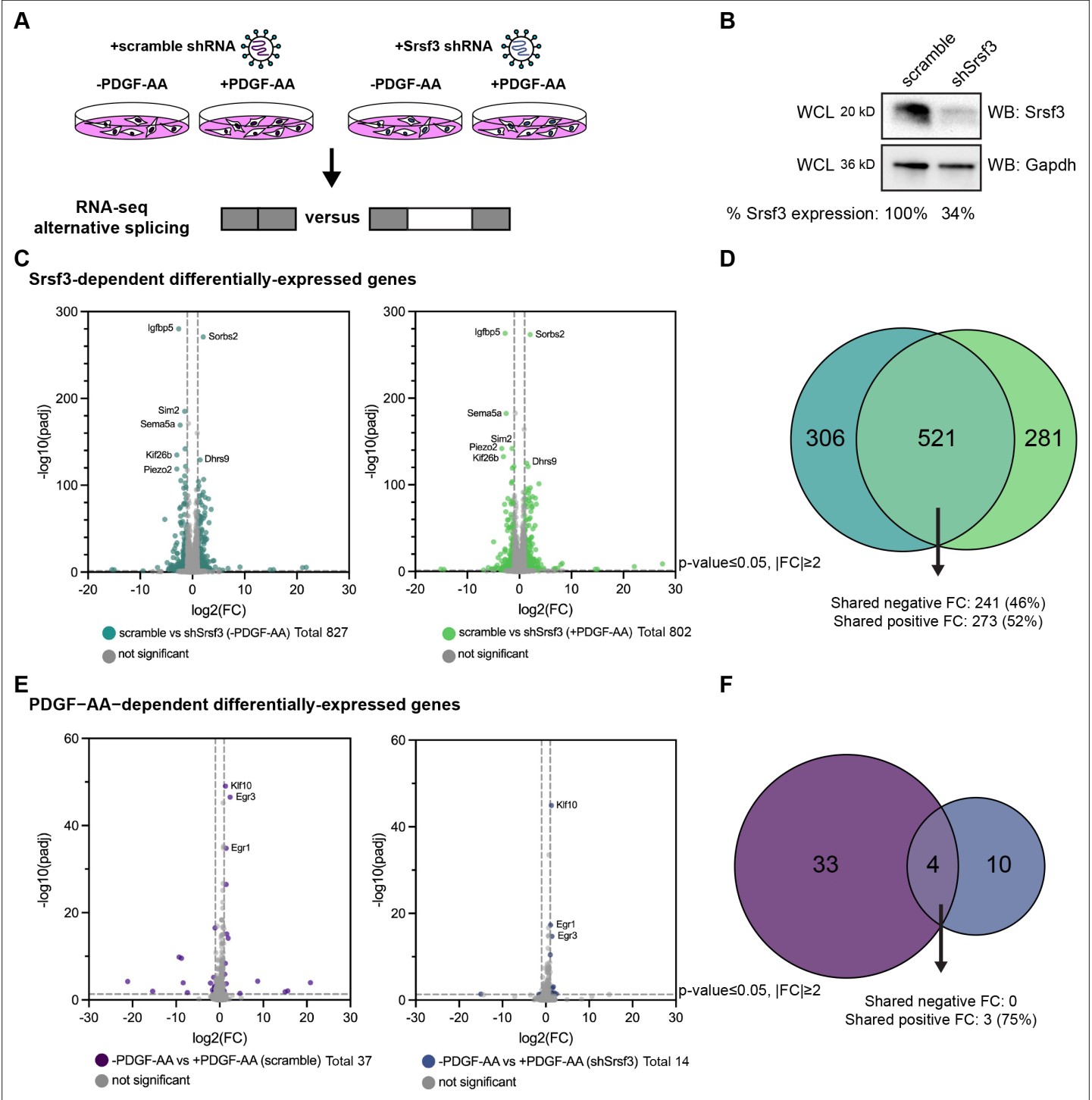

**Figure 1.** PDGFRα signaling for 1 hr minimally affects gene expression. (**A**) Schematic of RNA-seq experimental design. Immortalized mouse embryonic palatal mesenchyme (iMEPM) cells were transduced to stably express a scramble shRNA (scramble) or shRNA targeting the 3' UTR of *Srsf3* (shSrsf3). iMEPM cells expressing either scramble or shSrsf3 were left unstimulated or stimulated with 10 ng/mL PDGF-AA for 1 hr and RNA was isolated for RNA-seq analysis. (**B**) Western blot (WB) analysis of whole-cell lysates (WCL) from scramble and shSrsf3 cell lines with anti-Srsf3 and anti-Gapdh antibodies. The percentage of Srsf3 expression normalized to Gapdh expression is indicated below. (**C**) Volcano plots depicting differentially expressed genes in scramble versus shSrsf3 cell lines in the absence (left) or presence (right) of PDGF-AA stimulation. Log$_2$ (fold change) (FC) values represent log$_2$ (shSrsf3 normalized counts/scramble normalized counts). Significant changes in gene-level expression are reported for genes with adjusted p (padj)<0.05 and fold change |FC| ≥ 2. (**D**) Venn diagram of significant genes from (**C**). (**E**) Volcano plots depicting differentially expressed genes in the absence versus

*Figure 1 continued on next page*

*Figure 1 continued*

presence of PDGF-AA ligand in scramble (left) or shSrsf3 (right) cell lines. Log$_2$ (FC) values represent log$_2$ (+PDGF-AA normalized counts/-PDGF-AA normalized counts). (**F**) Venn diagram of significant genes from (**E**).

The online version of this article includes the following source data and figure supplement(s) for figure 1:

**Source data 1.** Srsf3 western blots.

**Source data 2.** Gapdh western blots.

**Source data 3.** DESeq2 output.

**Figure supplement 1.** High correlation of Srsf3-dependent differentially expressed genes across ligand treatment conditions.

**Figure supplement 2.** Gene ontology (GO) analysis of differentially expressed genes across treatment comparisons.

## PDGFRα signaling for 1 hr has a more pronounced effect on alternative RNA splicing

We previously demonstrated that AS is an important mechanism of gene expression regulation downstream of PI3K/Akt-mediated PDGFRα signaling in the murine mid-gestation palatal shelves (*Dennison et al., 2021*). Accordingly, we next assessed AS in our same RNA-seq dataset. In examining Srsf3-dependent alternatively spliced transcripts, we detected 1354 differential AS events in the absence of ligand treatment and 1071 differential AS events upon ligand stimulation (*Figure 2A*). When filtered to include events detected in at least 10 reads in either condition, we obtained a list of 1113 differential AS events in the absence of ligand treatment and 795 differential AS events upon ligand stimulation (*Figure 2B*). There was limited overlap (203 out of 1705; 11.9%) of Srsf3-dependent alternatively spliced transcripts across ligand treatment conditions, resulting in a total of 1705 unique events within both datasets (*Figure 2A and B*). Of the 203 shared alternatively spliced transcripts, 100% had the same directionality, including 81 (40%) with shared negative changes in percent spliced in (PSI) (exon included more often in shSrsf3 samples) and 122 (60%) with shared positive changes in PSI (exon skipped more often in shSrsf3 samples) (*Figure 2A and B*). We confirmed the differential AS of two transcripts, *Arhgap12* and *Cep55*, between scramble and shSrsf3 samples by qPCR using primers within constitutively expressed exons flanking the alternatively spliced exon (*Figure 2—figure supplement 1A and B*). These findings demonstrate that AS of a set of transcripts (203) depends on Srsf3 activity independent of PDGFRα signaling, while a much larger set of transcripts (1502) is alternatively spliced in response to both Srsf3 activity and PDGFRα signaling (*Figure 2A and B*). GO analysis of these Srsf3-dependent alternatively spliced transcripts revealed that the most significant terms for biological process and molecular function commonly involved regulation of RNA splicing, and RNA binding and cadherin binding, respectively (*Figure 2—figure supplement 2A and B*).

In examining PDGF-AA-dependent alternatively spliced transcripts, we detected 595 differential AS events in scramble cells and 398 differential AS events in shSrsf3 cells (*Figure 2C*). When filtered to include events detected in at least 10 reads in either condition, we obtained a list of 375 differential AS events in scramble cells and 256 differential AS events in shSrsf3 cells (*Figure 2D*). There was negligible overlap (9 out of 622; 1.45%) of PDGF-AA-dependent alternatively spliced transcripts across Srsf3 conditions, resulting in a total of 622 unique events within both datasets (*Figure 2C and D*). Of the nine shared alternatively spliced transcripts, 100% had the same directionality, including five (56%) with shared negative changes in PSI (exon included more often in +PDGF-AA samples) and four (44%) with shared positive changes in PSI (exon skipped more often in +PDGF-AA samples) (*Figure 2C and D*). Taken together, these findings demonstrate that both Srsf3 activity and PDGFRα signaling have more pronounced effects on AS than gene expression. While more transcripts and genes are subject to these regulatory mechanisms upon loss of Srsf3 activity, the magnitude of events is more greatly skewed toward AS upon PDGF-AA stimulation. Further, these results show that AS of a set of transcripts (9) depends on PDGFRα signaling independent of Srsf3 activity, while a much larger set of transcripts (613) is alternatively spliced in response to both PDGFRα signaling and Srsf3 activity (*Figure 2C and D*). When combined with the data above (*Figure 2B*), this points to a profound dependence between PDGFRα signaling and Srsf3 in regulating AS of transcripts in the facial mesenchyme. GO analysis of these PDGF-AA-dependent alternatively spliced transcripts demonstrated only a handful of significant terms for molecular function in the scramble cells, all relating to protein kinase activity (*Figure 2—figure supplement 2B*).

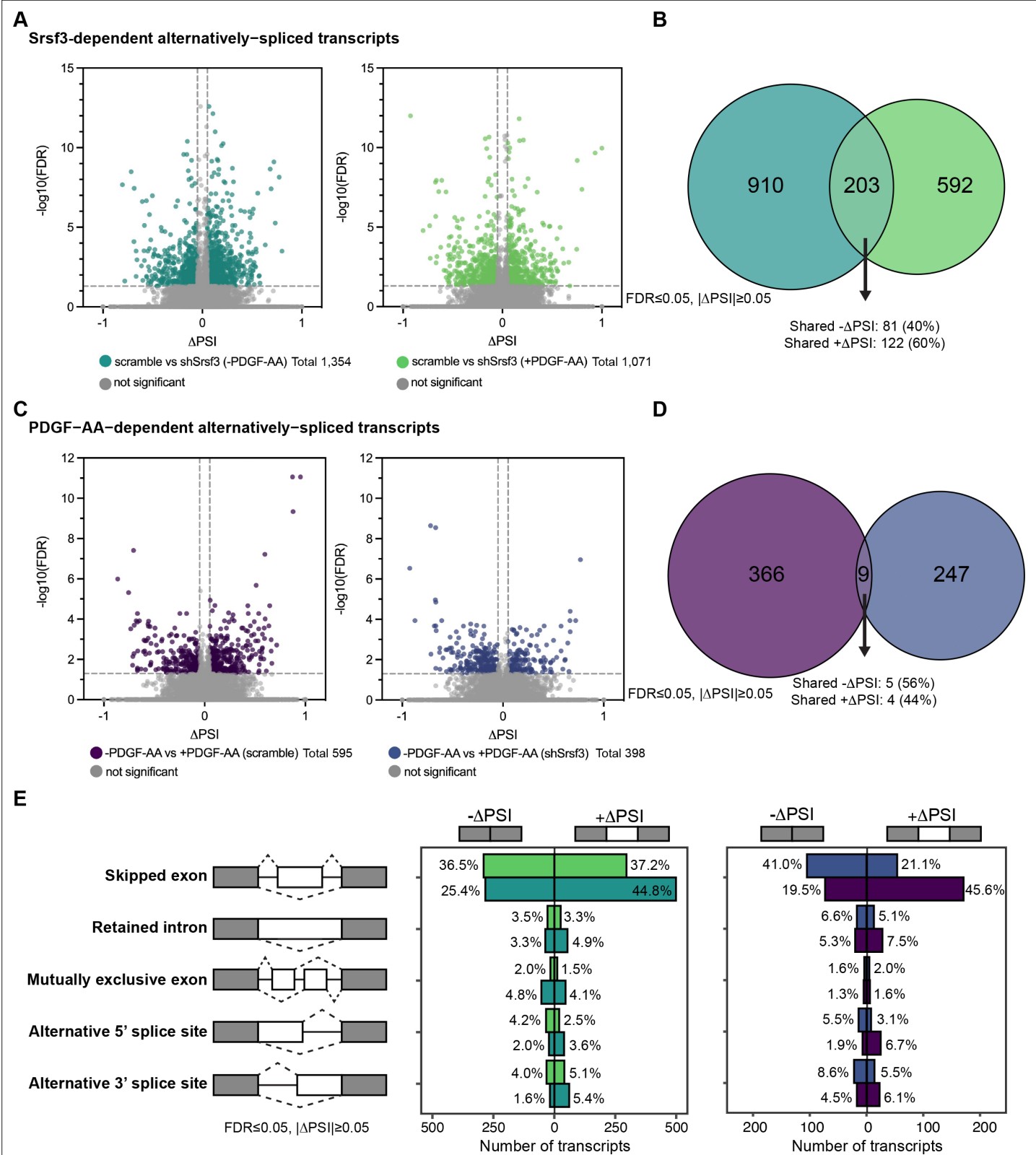

**Figure 2.** PDGFRα signaling for 1 hr has a more pronounced effect on alternative RNA splicing. (**A**) Volcano plots depicting alternatively spliced transcripts in scramble versus shSrsf3 cell lines in the absence (left) or presence (right) of PDGF-AA stimulation. Difference in percent spliced in (ΔPSI) values represent scramble PSI – shSrsf3 PSI. Significant changes in alternative RNA splicing are reported for events with a false discovery rate (FDR) ≤ 0.05 and a difference in percent spliced in (|ΔPSI|) ≥ 0.05. (**B**) Venn diagram of significant transcripts from (**A**), filtered to include events detected in

*Figure 2 continued on next page*

*Figure 2 continued*

at least 10 reads in either condition. (**C**) Volcano plots depicting alternatively spliced transcripts in the absence versus presence of PDGF-AA ligand in scramble (left) or shSrsf3 (right) cell lines. Difference in percent spliced in (ΔPSI) values represent -PDGF-AA PSI –+PDGF-AA PSI. (**D**) Venn diagram of significant transcripts from (**C**), filtered to include events detected in at least 10 reads in either condition. (**E**) Bar graphs depicting alternative RNA splicing events in scramble versus shSrsf3 cell lines in the absence or presence of PDGF-AA stimulation (left) or in the absence versus presence of PDGF-AA ligand in scramble or shSrsf3 cell lines (right).

The online version of this article includes the following source data and figure supplement(s) for figure 2:

**Source data 1.** rMATS output for scramble (-PDGF-AA) versus shSrsf3 (-PDGF-AA) RNA-seq analysis.

**Source data 2.** rMATS output for scramble (+PDGF-AA) versus shSrsf3 (+PDGF-AA) RNA-seq analysis.

**Source data 3.** rMATS output for -PDGF-AA (scramble) versus +PDGF-AA (scramble) RNA-seq analysis.

**Source data 4.** rMATS output for -PDGF-AA (shSrsf3) versus +PDGF-AA (shSrsf3) RNA-seq analysis.

**Figure supplement 1.** qPCR validation of differential AS between scramble and shSrsf3 samples.

**Figure supplement 1—source data 1.** *Arhgap12* and *Cep55* PCR gels.

**Figure supplement 2.** Gene ontology (GO) analysis of alternatively spliced transcripts across treatment comparisons.

The vast majority of Srsf3-dependent (70.2–73.7%) and PDGF-AA-dependent (62.1–65.1%) AS events involved skipped exons, with minimal contribution from retained introns, mutually exclusive exons, alternative 5' splice sites, or alternative 3' splice sites (*Figure 2E*). For the Srsf3-dependent skipped exon events, there were more transcripts with +ΔPSI (exon skipped more often in shSrsf3 samples) (44.8%) as opposed to -ΔPSI (exon included more often in shSrsf3 samples) (25.4%) in the absence of PDGF-AA stimulation (*Figure 2E*), consistent with previous results that SR proteins tend to promote exon inclusion (*Fu and Ares, 2014*; *Licatalosi and Darnell, 2010*). However, PDGF-AA ligand treatment led to an increase in the percentage of transcripts with -ΔPSI (36.5%) (*Figure 2E*), indicating that PDGFRα signaling promotes exon skipping in the presence of Srsf3. Among the PDGF-AA-dependent skipped exon events, there was a significant shift in transcripts with -ΔPSI (exon included more often in +PDGF-AA samples) in the absence (41.0%) versus presence (19.5%) of Srsf3, and a corresponding shift in transcripts with +ΔPSI (exon skipped more often in +PDGF-AA samples) (21.1% and 45.6%, respectively) (*Figure 2E*), again suggesting that PDGF-AA stimulation causes increased exon skipping when Srsf3 is present.

## Srsf3 exhibits differential transcript binding upon PDGFRα signaling

To determine direct binding targets of Srsf3 downstream of PDGFRα signaling, we conducted eCLIP (*Van Nostrand et al., 2017*; *Van Nostrand et al., 2016*) of iMEPM cells that were left unstimulated (-PDGF-AA) or treated with 10 ng/mL PDGF-AA for 1 hr (+PDGF-AA) as above (*Figure 3A*). Immunoprecipitation with a previously validated Srsf3 antibody (*Dennison et al., 2021*) successfully enriched for Srsf3 in UV-crosslinked cells (*Figure 3B*). We detected 6555 total eCLIP peaks in protein-coding genes in the -PDGF-AA samples and 8584 total peaks in the +PDGF-AA samples. Among the -PDGF-AA peaks, 3727 (56.9%) were located in exons (CDS) and 1690 (25.8%) were located within introns, with the rest binding within 5' UTRs (768, 11.7%) and 3' UTRs (367, 5.60%) (*Figure 3C and D*). We observed substantial shifts in Srsf3-bound regions upon PDGF-AA stimulation. Many more +PDGF-AA peaks were located within exons (7139, 83.2%), and less were located within introns (765, 8.91%), 5' UTRs (389, 4.53%), and 3' UTRs (287, 3.34%) (*Figure 3C and D*). Given that SR proteins are crucial for exon definition and bind to exonic splicing enhancers to recruit and stabilize core splicing machinery (*Fu and Maniatis, 1990*; *Krainer et al., 1991*; *Zahler et al., 1993*), we investigated Srsf3 binding around 5' and 3' splice sites in response to PDGFRα signaling. Consistent with the results above and previous findings (*Änkö et al., 2012*), Srsf3 binding was enriched in exonic regions, as opposed to intronic regions, surrounding the splice sites (*Figure 3E*). There was greater mean coverage of Srsf3 peaks in the +PDGF-AA condition versus the -PDGF-AA condition within 100 nucleotides upstream of the 5' splice site and within 100 nucleotides downstream of the 3' splice site (*Figure 3E*). Additionally, we detected decreased mean coverage in the +PDGF-AA condition within 25 nucleotides downstream of the 5' splice site boundary (*Figure 3E*). Taken together, these data show that PDGFRα signaling leads to increased binding of Srsf3 to exons. Finally, we performed unbiased motif enrichment analysis of Srsf3 peaks in unstimulated and PDGF-AA-treated samples, revealing that the mostly highly enriched motifs in -PDGF-AA samples were CACACA and AAGAAG

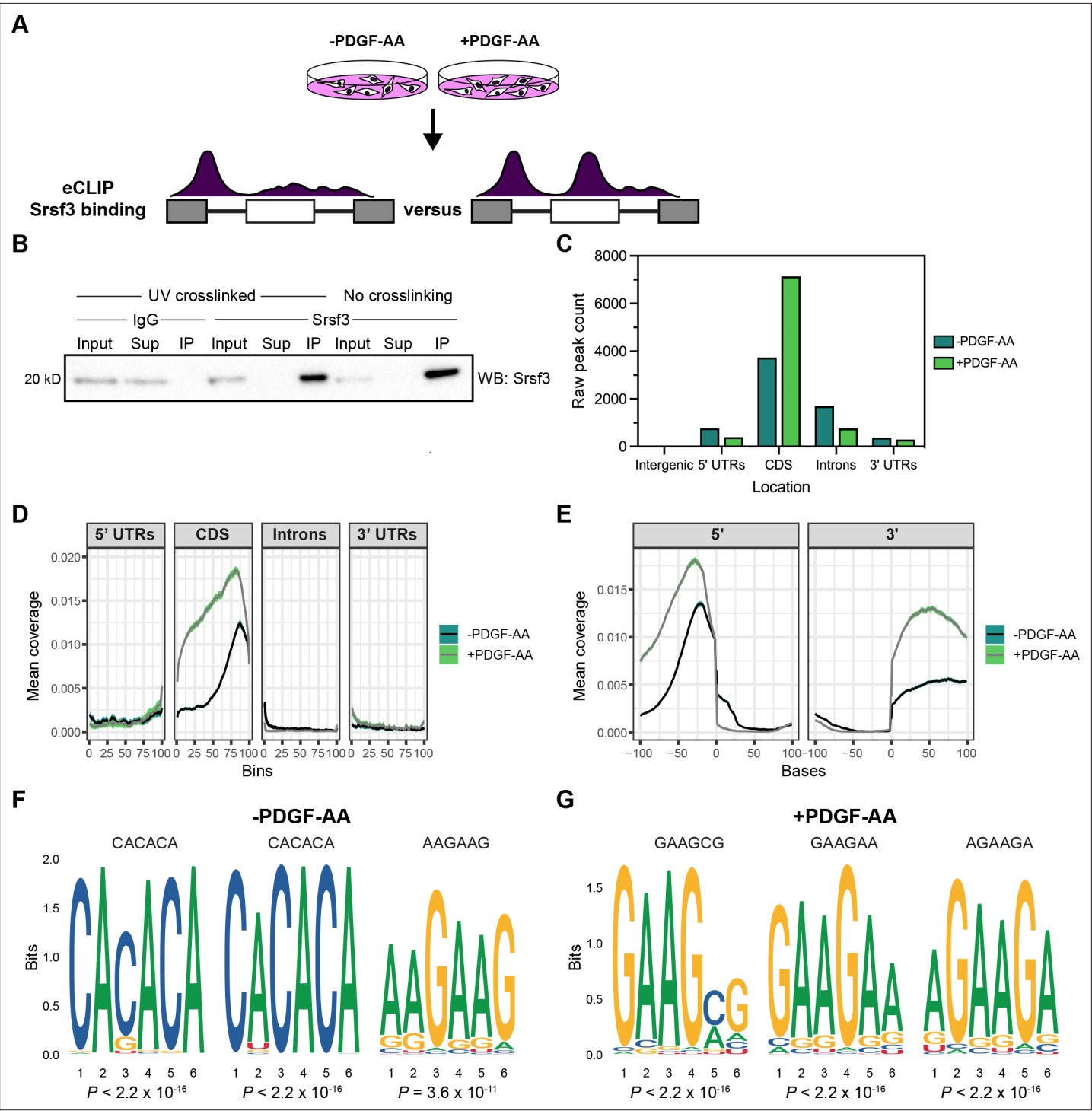

**Figure 3.** Srsf3 exhibits differential transcript binding upon PDGFRα signaling. (**A**) Schematic of enhanced UV-crosslinking and immunoprecipitation (eCLIP) experimental design. Immortalized mouse embryonic palatal mesenchyme (iMEPM) cells were left unstimulated or stimulated with 10 ng/mL PDGF-AA for 1 hr and processed for eCLIP analysis. (**B**) Immunoprecipitation (IP) of Srsf3 from cells that were UV-crosslinked or not UV-crosslinked with IgG or an anti-Srsf3 antibody followed by western blotting (WB) of input, supernatant (Sup), and IP samples with an anti-Srsf3 antibody. (**C**) Mapping of eCLIP peaks to various transcript locations in the absence or presence of PDGF-AA stimulation. 5′ UTR, 5′ untranslated region; CDS, coding sequence; 3′ UTR, 3′ untranslated region. (**D, E**) Mean coverage of eCLIP peaks across various transcript locations (**D**) and surrounding the 5′ and 3′ splice sites (**E**) in the absence or presence of PDGF-AA stimulation. (**F, G**) Top three motifs enriched in eCLIP peaks in the absence (**F**) or presence (**G**) of PDGF-AA stimulation with associated p-values (t-test).

The online version of this article includes the following source data and figure supplement(s) for figure 3:

*Figure 3 continued on next page*

*Figure 3 continued*

**Source data 1.** Srsf3 western blots.

**Source data 2.** eCLIP output.

**Source data 3.** Raw peak counts of eCLIP peaks across various transcript locations.

**Figure supplement 1.** PDGFRα signaling influences Srsf3 binding specificity.

(*Figure 3F*, *Figure 3—figure supplement 1*). Of note, these CA-rich motifs have previously been identified as canonical Srsf3 motifs in a CLIP study utilizing a stably integrated SRSF3-EGFP transgene (*Änkö et al., 2012*). However, in PDGF-AA-stimulated samples, the most highly enriched motifs were GAAGCG, GAAGAA, and AGAAGA (*Figure 3G*, *Figure 3—figure supplement 1*), suggesting that PDGFRα signaling influences Srsf3 binding specificity.

## Srsf3 and PDGFRα signaling are associated with differential GC content and length of alternatively spliced exons

We next probed our RNA-seq dataset to determine whether specific transcript features were associated with Srsf3-dependent AS. When comparing significant AS events between scramble and shSrsf3 samples in the absence of PDGF-AA stimulation, we found that included exons had a significantly higher GC content (median value of 53.4%) than both skipped exons (50.0%) and exons that were not differentially alternatively spliced (51.2%) when Srsf3 is present (*Figure 4A*). Additionally, we observed that included exons had a significantly lower ratio of downstream intron to exon GC content (0.856) than both skipped exons (0.901) and exons that were not differentially alternatively spliced (0.888) in the presence of Srsf3 (*Figure 4B*). The same comparisons revealed that the ratio of upstream and downstream intron to exon length was significantly decreased in included exons (median values of 12.4 and 10.6, respectively) compared to both skipped exons (19.5 and 20.0) and exons that were not differentially alternatively spliced (14.2 and 13.3) in the presence of Srsf3 (*Figure 4C and D*). Taken together, our data demonstrate that exons that are included in the presence of Srsf3 tend to have a higher GC content and lower intron to exon length ratio.

To determine whether PDGFRα signaling had an effect on the transcript features to which Srsf3 bound, we subsequently examined our eCLIP dataset. We found that the exon GC content was significantly increased in exons bound by Srsf3 in the absence of ligand treatment (median value of 57.9%) compared to unbound exons (51.5%) (*Figure 4E*). However, exon GC content was similar between unbound exons and those bound by Srsf3 upon PDGF-AA stimulation (53.8%) (*Figure 4E*). These findings indicate that PDGFRα signaling mediates binding of Srsf3 to exons with a lower GC content.

## Transcripts bound by Srsf3 that undergo alternative splicing upon PDGFRα signaling encode regulators of PI3K signaling

To determine which transcripts are directly bound by Srsf3 and subject to DE and/or AS, we cross-referenced the eCLIP and RNA-seq datasets. We collated transcripts with an Srsf3 eCLIP peak that were uniquely detected in the -PDGF-AA or +PDGF-AA samples (2660 transcripts across 3388 peaks). Similarly, we gathered DE genes (596) or differentially alternatively spliced transcripts (985) uniquely found in one of the four treatment comparisons (*Figure 5A*). Only 32 (5.4%) of the DE genes were directly bound by Srsf3, while 233 (23.7%) of the alternatively spliced transcripts had an Srsf3 eCLIP peak, with very little overlap (one transcript) between all three categories (*Figure 5A*).

We next sought to identify high-confidence transcripts for which Srsf3 binding had an increased likelihood of contributing to AS. Previous studies revealed enrichment of functional RBP motifs near alternatively spliced exons (*Yee et al., 2019*). As such, we correlated the eCLIP peaks with AS events across all four treatment comparisons by identifying transcripts in which Srsf3 bound within an alternatively spliced exon or within 250 bp of the neighboring introns. In agreement with our findings above for the entire eCLIP dataset, Srsf3 exhibited differential binding in exons and surrounding the 5' and 3' splice sites upon PDGF-AA stimulation in these high-confidence, overlapping datasets (*Figure 5—figure supplement 1A and B*). We performed an unbiased motif enrichment analysis of Srsf3 peaks within the high-confidence, overlapping datasets, again revealing different motifs between ligand treatment conditions and an enrichment of GA-rich motifs in the +PDGF-AA samples (*Figure 5—figure supplement 1C and D*).

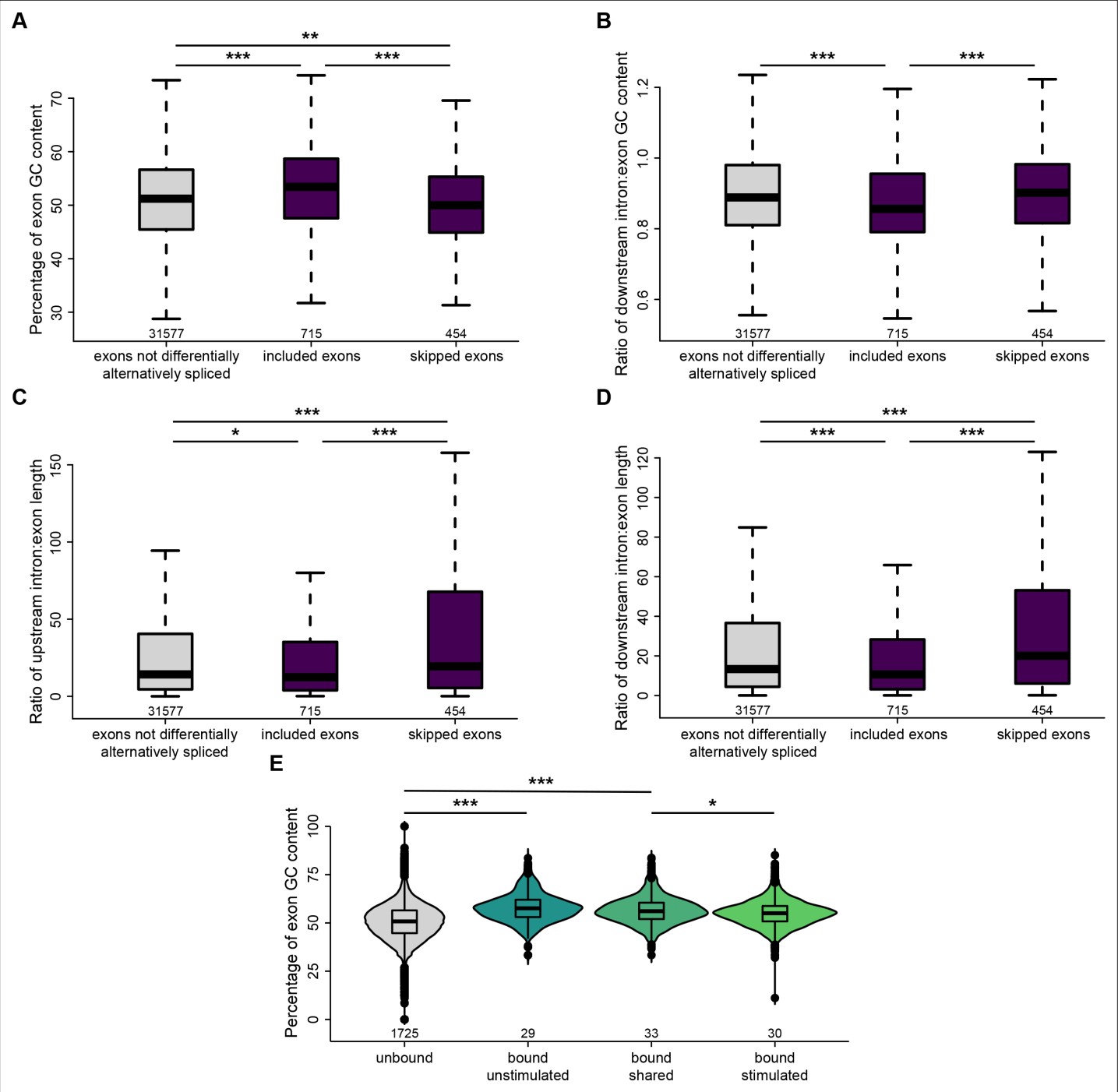

**Figure 4.** Srsf3 and PDGFRα signaling are associated with differential GC content and length of alternatively spliced exons. (**A**) Box and whisker plot depicting the percentage of exon GC content in exons that are not differentially alternatively spliced, and exons that are included or skipped when Srsf3 is present from the rMATS analysis. (**B**) Box and whisker plot depicting the ratio of downstream intron to exon GC content in exons that are not differentially alternatively spliced, and exons that are included or skipped when Srsf3 is present from the rMATS analysis. (**C, D**) Box and whisker plots depicting the ratio of upstream intron to exon length (**C**) and downstream intron to exon length (**D**) in exons that are not differentially alternatively spliced, and exons that are included or skipped when Srsf3 is present from the rMATS analysis. (**E**) Violin and box and whisker (inset) plots depicting the percentage of exon GC content in exons that are not bound by Srsf3, and exons that are bound in the absence and/or presence of PDGF-AA stimulation from the enhanced UV-crosslinking and immunoprecipitation (eCLIP) analysis. *p<0.05; **p<0.01; ***p<0.001 (Mann–Whitney *U*-test).

The online version of this article includes the following source data for figure 4:

**Source data 1.** Matt features.

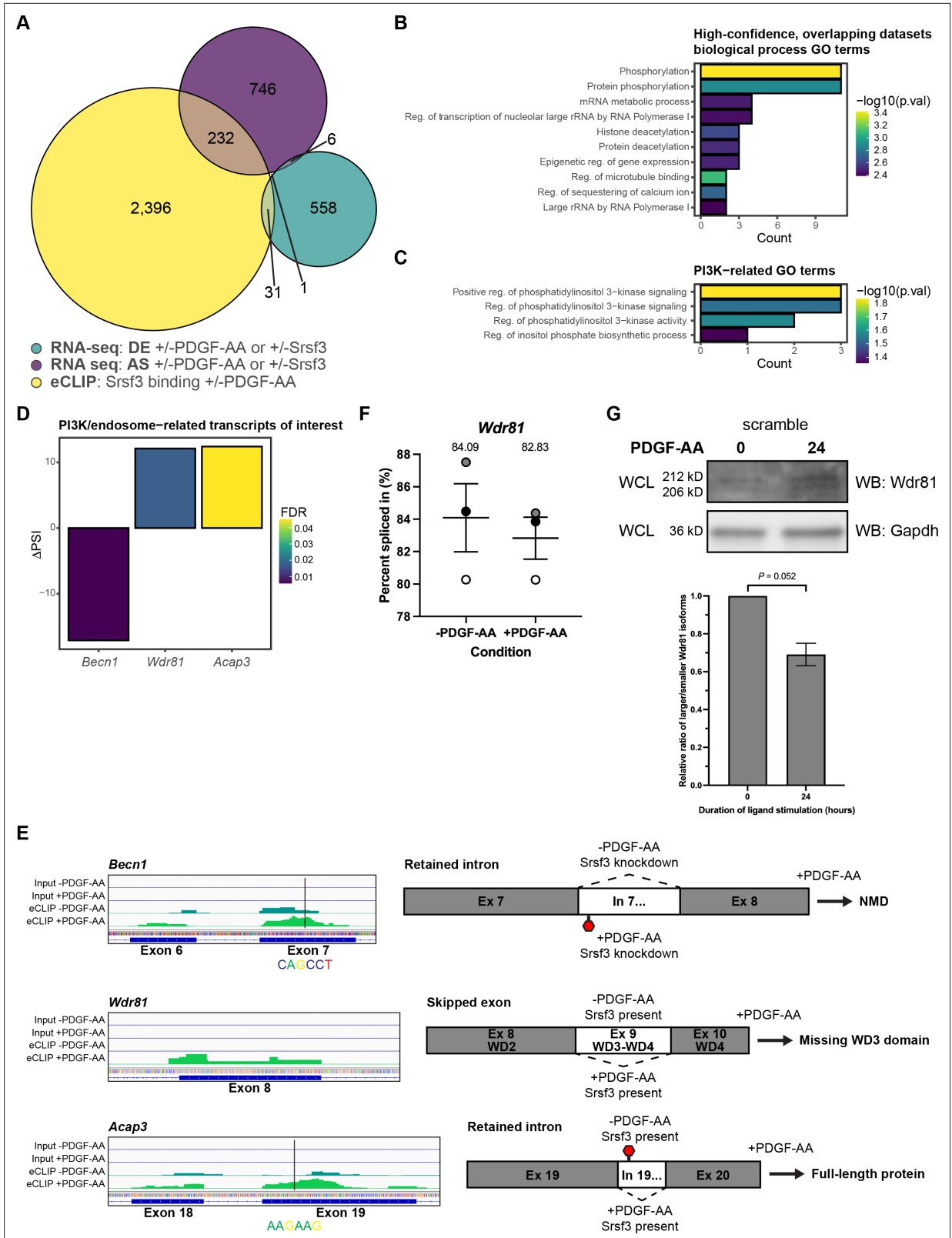

**Figure 5.** Transcripts bound by Srsf3 that undergo alternative splicing upon PDGFRα signaling encode regulators of PI3K signaling. (**A**) Venn diagram of genes with differential expression (DE) or transcripts subject to alternative RNA splicing (AS) across the four treatment comparisons that overlap with transcripts with Srsf3 enhanced UV-crosslinking and immunoprecipitation (eCLIP) peaks in the absence or presence of PDGF-AA stimulation. (**B, C**) Top 10 (**B**) and PI3K-related (**C**) biological process gene ontology (GO) terms for transcripts from the high-confidence, overlapping datasets. p.val,

*Figure 5 continued on next page*

*Figure 5 continued*

p. (**D**) Difference in percent spliced in (ΔPSI) values for PI3K/endosome-related transcripts of interest. ΔPSI values represent -PDGF-AA PSI – +PDGF-AA PSI. FDR, false detection rate. (**E**) Peak visualization for input and eCLIP samples in the absence or presence of PDGF-AA stimulation from Integrative Genomics Viewer (left) with location of motifs from *Figure 5—figure supplement 1* indicated below for PI3K/endosome-related transcripts of interest. Predicted alternative splicing outcomes for PI3K/endosome-related transcripts of interest (right). (**F**) Scatter dot plot depicting the percent spliced in as assessed by qPCR analysis of *Wdr81* exon 9 in the scramble cell line following 1 hr of PDGF-AA stimulation quantified from n = 3 biological replicates. Data are mean ± s.e.m. Shaded circles correspond to independent experiments. (**G**) Western blot (WB) analysis of whole-cell lysates (WCL) from the scramble cell line following 24 hr of PDGF-AA stimulation with anti-Wdr81 and anti-Gapdh antibodies. Bar graph depicting relative ratios of larger/smaller Wdr81 isoforms quantified from n = 3 biological replicates as above. Data are mean ± s.e.m. p=0.052 (two-tailed, ratio paired *t*-test).

The online version of this article includes the following source data and figure supplement(s) for figure 5:

**Source data 1.** List of transcripts and genes from Venn diagram in *Figure 5A*.

**Source data 2.** High-confidence, overlapping dataset output correlating eCLIP with scramble (-PDGF-AA) versus shSrsf3 (-PDGF-AA) rMATS RNA-seq analysis.

**Source data 3.** High-confidence, overlapping dataset output correlating eCLIP with scramble (+PDGF-AA) versus shSrsf3 (+PDGF-AA) rMATS RNA-seq analysis.

**Source data 4.** High-confidence, overlapping dataset output correlating eCLIP with -PDGF-AA (scramble) versus +PDGF-AA (scramble) rMATS RNA-seq analysis.

**Source data 5.** High-confidence, overlapping dataset output correlating eCLIP with -PDGF-AA (shSrsf3) versus +PDGF-AA (shSrsf3) rMATS RNA-seq analysis.

**Source data 6.** Wdr81 western blots.

**Source data 7.** Gapdh western blots.

**Source data 8.** Relative ratios of larger/smaller Wdr81 isoforms.

**Figure supplement 1.** Srsf3 exhibits differential transcript binding upon PDGFRα signaling in the subset of transcripts from the high-confidence, overlapping datasets.

To determine whether transcripts that are differentially bound by Srsf3 and undergo differential AS downstream of PDGFRα signaling contribute to shared biological outputs, we conducted GO analysis of the 149 unique transcripts from the high-confidence, overlapping datasets. The most significant terms for biological process involved protein phosphorylation and deacetylation, and RNA metabolism (*Figure 5B*). Given that PI3K-mediated PDGFRα signaling is critical for craniofacial development and regulates AS in this context (*Dennison et al., 2021*; *Fantauzzo and Soriano, 2014*; *Klinghoffer et al., 2002*), we turned our focus toward GO terms associated with this signaling pathway. We noted enrichment of PI3K-related GO terms (*Figure 5C*), which encompassed the genes *Becn1*, *Wdr81*, and *Acap3* (*Figure 5D*). Related to their roles in PI3K signaling, each of these gene products has been shown to regulate membrane and/or endocytic trafficking. Phosphatidylinositol 3-phosphate (PI(3)P) is a critical component of early endosomes and is mainly generated by conversion of phosphatidylinositol (PI) by the class III PI3K complex, which includes Beclin-1 (encoded by *Becn1*) (*Wallroth and Haucke, 2018*). WDR81 and Beclin-1 have been shown to interact, resulting in decreased endosomal PI(3)P synthesis, PI(3)P turnover, and early endosome conversion to late endosomes (*Liu et al., 2016*). Importantly, this role of WDR81 contributes to RTK degradation (*Rapiteanu et al., 2016*). Finally, Acap3 is a GTPase-activating protein (GAP) for the small GTPase Arf6, converting Arf6 to an inactive, GDP-bound state (*Miura et al., 2016*). Arf6 localizes to the plasma membrane and endosomes, and has been shown to regulate endocytic membrane trafficking by increasing PI(4,5)P2 levels at the cell periphery (*D'Souza-Schorey and Chavrier, 2006*). Further, constitutive activation of Arf6 leads to upregulation of the gene encoding the p85 regulatory subunit of PI3K and increased activity of both PI3K and AKT (*Yoo et al., 2019*).

Within our data, Srsf3 binding was increased in *Becn1* exon 7 upon PDGF-AA stimulation, at an enriched motif within the high-confidence, overlapping datasets, and we observed a corresponding increase in retention of adjacent intron 7 (*Figure 5D and E*). As *Becn1* intron 7 contains a premature termination codon (PTC), this event is predicted to result in nonsense-mediated decay (NMD) in the absence of Srsf3 (*Figure 5E*). Srsf3 binding was also increased in *Wdr81* exon 8 in response to PDGF-AA treatment, and our analyses revealed a corresponding increase in excision of adjacent exon 9 (*Figure 5D and E*). Because *Wdr81* exon 9 encodes two WD-repeat domains, which are generally believed to form a β-propeller structure required for protein interactions (*Li and Roberts, 2001*), this

AS event is predicted to result in a protein missing a functional domain (*Figure 5E*). These splicing patterns predict increased levels of Beclin-1 and decreased levels of functional Wdr81 in the presence of Srsf3 and PDGF-AA stimulation, resulting in augmented early endosome formation. Srsf3 binding was additionally increased in *Acap3* exon 19 upon PDGF-AA stimulation, at an enriched motif within the high-confidence, overlapping datasets, and we observed a corresponding increase in excision of adjacent intron 19 (*Figure 5D and E*). As *Acap3* intron 19 contains a PTC, this event is predicted to result in more transcripts encoding full-length protein (*Figure 5E*).

Finally, as Wdr81 protein levels are predicted to regulate RTK trafficking between early and late endosomes, we confirmed the differential AS of *Wdr81* transcripts between unstimulated scramble cells and scramble cells treated with PDGF-AA ligand for 1 hr by qPCR using primers within constitutively expressed exons flanking alternatively spliced exon 9. This analysis revealed a decreased PSI for *Wdr81* in each of three biological replicates upon PDGF-AA ligand treatment (*Figure 5F*). Relatedly, we assessed the ratio of larger isoforms of Wdr81 protein (containing the WD3 domain) to smaller isoforms (missing the WD3 domain) via western blotting. Consistent with our RNA-seq and qPCR results, PDGF-AA stimulation for 24 hr in the presence of Srsf3 led to an increase in smaller Wdr81 protein isoforms (*Figure 5G*).

## Srsf3 regulates early endosome size and phosphorylation of Akt downstream of PDGFRα signaling

Given that Srsf3 differentially binds to transcripts that encode proteins involved in early endosomal trafficking downstream of PDGFRα signaling, we first examined the formation of Rab5-positive early endosomes (*Zerial and McBride, 2001*) in response to a time course of PDGF-AA ligand stimulation in scramble versus shSrsf3 cells. As expected, Rab5 puncta size increased from 0 min ($9.79 \times 10^{-4} \pm 1.13 \times 10^{-4}$ arbitrary units; mean ± s.e.m.) to 15 min of ligand stimulation ($2.01 \times 10^{-3} \pm 7.20 \times 10^{-4}$ arbitrary units) in scramble cells, and significantly so by 60 min ($2.58 \times 10^{-3} \pm 9.20 \times 10^{-4}$ arbitrary units) (*Figure 6A, C–C‴, E–E‴ and G–G‴*). However, this increase was not observed in the absence of Srsf3 ($9.21 \times 10^{-4} \pm 1.61 \times 10^{-4}$ arbitrary units at 60 min) (*Figure 6A, D–D‴, F–F‴ and H–H‴*), demonstrating that Srsf3-mediated PDGFRα signaling leads to enlarged early endosomes.

We next examined colocalization of PDGFRα with Rab5, as an estimate of receptor levels in early endosomes. Colocalization levels increased from 0 min (0.332 ± 0.0832 Pearson's correlation coefficient [PCC]; mean ± s.e.m.) to 15 min (0.429 ± 0.108 PCC) of PDGF-AA treatment in scramble cells, then decreased to near baseline levels by 60 min (0.348 ± 0.0885 PCC) (*Figure 6B, C–C‴, E–E‴ and G–G‴*) as a subset of PDGFRα homodimers are likely trafficked to late endosomes (*Rogers et al., 2022*). Strikingly, shSrsf3 cells exhibited a significant decrease in colocalization between PDGFRα and Rab5 by 60 min of ligand treatment (0.186 ± 0.0102 PCC) (*Figure 6B, D–D‴, F–F‴ and H–H‴*), indicating that Srsf3 activity downstream of PDGFRα signaling results in retention of PDGFRα in early endosomes.

Finally, we previously demonstrated that rapid internalization of PDGFRα homodimers following PDGF-AA ligand stimulation is critical for downstream AKT phosphorylation (*Rogers et al., 2022*). As such, we examined phospho-Akt levels as a readout of PDGFRα activation in early endosomes. While PDGF-AA treatment for 15 min induced a robust phospho-Akt response in scramble cells (13.5 ± 7.39 relative induction, mean ± s.e.m.) this response was muted in shSrsf3 cells at the same timepoint (5.73 ± 1.91) (*Figure 6I*). These findings further suggest that retention of PDGFRα in early endosomes leads to increases in downstream PI3K-mediated Akt signaling. Collectively, our data point to a feedback loop in which PI3K/Akt-mediated PDGFRα signaling results in the nuclear translocation of Srsf3 and the subsequent AS of transcripts to decrease levels of proteins that promote PDGFRα trafficking out of early endosomes (*Figure 6J*).

## Discussion

In this study, we confirmed our prior in vivo results in the mouse embryonic facial mesenchyme (*Dennison et al., 2021*) that PDGFRα signaling primary regulates gene expression via AS. PDGF-AA-dependent differential gene expression was minimal following 1 hr of ligand treatment and led to increased expression of immediate early genes *Klf10*, *Egf3*, and *Egr1*, consistent with previous findings (*Vasudevan et al., 2015*) and in line with the enriched GO terms of regulation of transcription

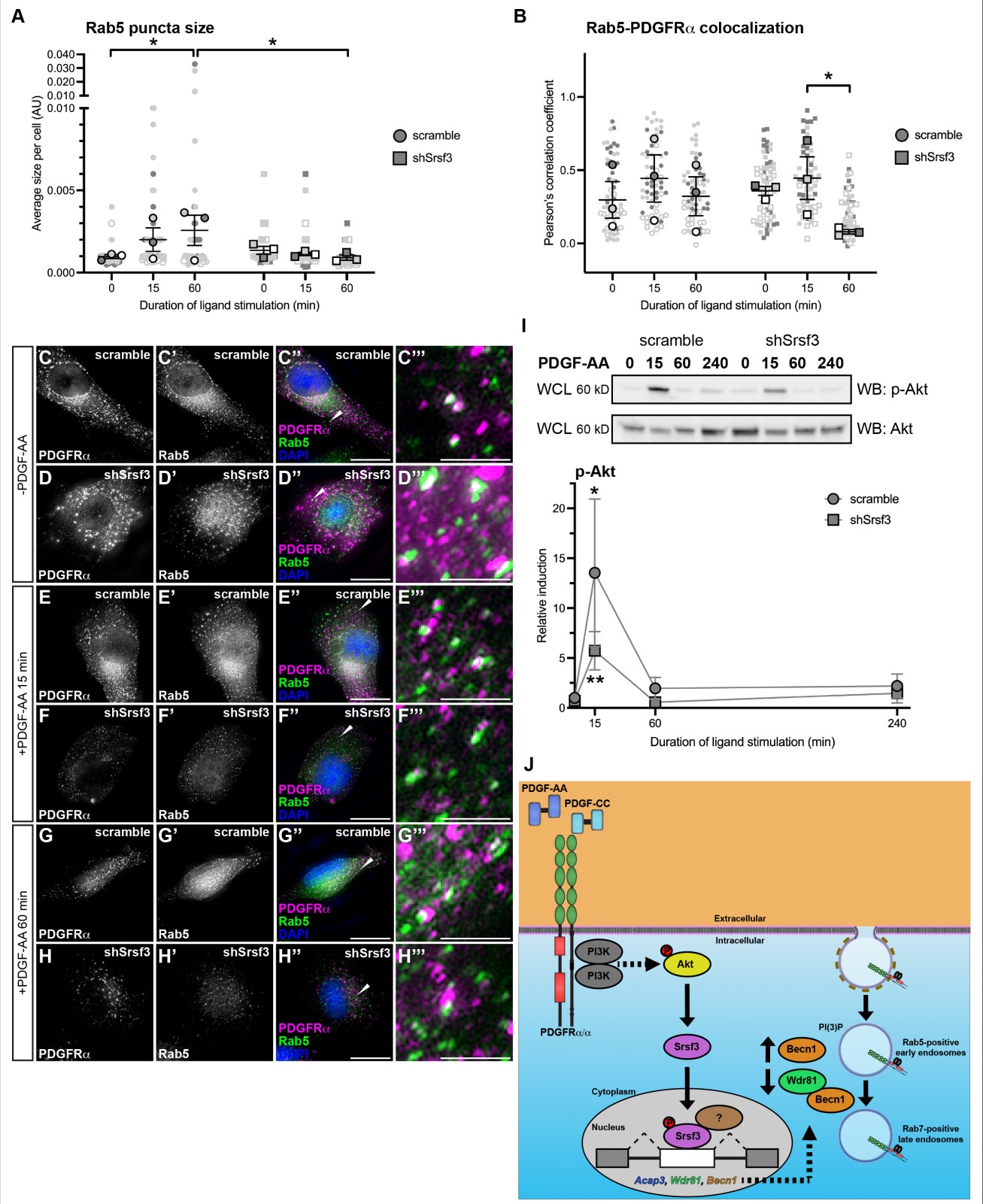

**Figure 6.** Srsf3 regulates early endosome size and phosphorylation of Akt downstream of PDGFRα signaling. (**A, B**) Scatter dot plots depicting average size of Rab5 puncta per cell (**A**) and Pearson's correlation coefficient of signals from anti-Rab5 and anti-PDGFRα antibodies (**B**) in scramble and shSrsf3 cell lines in the absence or presence (15–60 min) of PDGF-AA stimulation. Data are mean ± s.e.m. *p<0.05 (two-way ANOVA followed by uncorrected Fisher's LSD test). Shaded shapes correspond to independent experiments. Summary statistics from biological replicates consisting of independent

*Figure 6 continued on next page*

*Figure 6 continued*

experiments (large shapes) are superimposed on top of data from all cells; n = 20 technical replicates across each of three biological replicates. (**C–H'''**) PDGFRα antibody signal (white or magenta) and Rab5 antibody signal (white or green) as assessed by immunofluorescence analysis of scramble and shSrsf3 cells in the absence or presence (15–60 min) of PDGF-AA stimulation. Nuclei were stained with DAPI (blue). White arrows denote regions of colocalization, which are expanded in (**C''''–H''''**). Scale bars: 20 μm (**C–H''**), 3 μm (**C''''–H'''**). (**I**) Western blot (WB) analysis of whole-cell lysates (WCL) from scramble (left) and shSrsf3 (right) cell lines following a time course of PDGF-AA stimulation from 15 min to 4 hr, with anti-phospho-(p)-Akt and anti-Akt antibodies. Line graph depicting quantification of band intensities from n = 4 biological replicates as above. Data are mean ± s.e.m. *p<0.05; **p<0.01 (two-tailed, ratio paired *t*-test within each cell line and a two-tailed, unpaired *t*-test with Welch's correction between each cell line). (**J**) Model of experimental results in which PI3K/Akt-mediated PDGFRα signaling results in the nuclear translocation of Srsf3 and the subsequent AS of transcripts to decrease levels of proteins that promote PDGFRα trafficking out of early endosomes.

The online version of this article includes the following source data for figure 6:

**Source data 1.** Phospho-Akt western blots.

**Source data 2.** Akt western blots.

**Source data 3.** Relative induction values of phospho-Akt.

and DNA binding. Alternatively, PDGF-AA-dependent alternatively spliced transcripts were enriched for protein kinase activity, consistent with our prior AS findings upon disruption of PI3K-mediated PDGFRα signaling in the palatal shelf mesenchyme (*Dennison et al., 2021*). Importantly, our results demonstrated a significant dependence on the RBP Srsf3 for AS downstream of PDGFRα activation. In fact, we found that 88% of Srsf3-dependent and 99% of PDGF-AA-dependent alternatively spliced transcripts were responsive to both Srsf3 activity and PDGFRα signaling. As discussed above, Srsf3 is phosphorylated downstream of multiple stimuli (*Bavelloni et al., 2014*; *Dennison et al., 2021*; *Fantauzzo and Soriano, 2014*; *Long et al., 2019*; *Zhou et al., 2012*), and it is likely that these additional inputs contributed to Srsf3-dependent AS that was non-responsive to PDGF-AA treatment. Further, our previous mass spectrometry screen identified 11 additional RBPs involved in AS that are phosphorylated by Akt downstream of PI3K-mediated PDGFRα signaling in primary MEPM cells (*Fantauzzo and Soriano, 2014*), which may account for the small fraction of PDGF-AA-dependent AS that was non-responsive to Srsf3 knockdown. However, our RNA-seq results together with the phenotypic overlap of embryos with neural crest-specific ablation of *Srsf3* and mutant mouse models of *Pdgfra* and/or its ligands (*Andrae et al., 2016*; *Dennison et al., 2021*; *Ding et al., 2004*; *Fantauzzo and Soriano, 2014*; *Fredriksson et al., 2012*; *Klinghoffer et al., 2002*; *Soriano, 1997*) make a compelling case for Srsf3 serving as a critical effector of PDGFRα signaling in the facial mesenchyme.

Here, we observed that Srsf3-dependent skipped exon events were enriched for transcripts with a +ΔPSI (exon skipped more often in shSrsf3 samples) in the absence of PDGF-AA stimulation, consistent with Srsf3 promoting exon inclusion. Alternatively, PDGF-AA ligand treatment led to an increase in the percentage of Srsf3-dependent transcripts with a -ΔPSI (exon included more often in shSrsf3 samples), suggesting that PDGFRα signaling causes decreased exon inclusion in the presence of Srsf3. Interestingly, a recent paper found that tethering Srsf3 downstream of the 5' splice site or upstream of the 3' splice site using MS2 stem loops did not lead to AS of a splicing reporter (*Schmok et al., 2024*). However, the assay did not test tethering within the exon and used a single minigene sequence context, and thus had the potential to lead to false negative results (*Schmok et al., 2024*). Whether phosphorylation of Srsf3 directly influences its binding to target RNAs or acts to modulate Srsf3 protein–protein interactions which then contribute to differential RNA binding remains to be determined, though findings from *Schmok et al., 2024* may argue for the latter mechanism. Studies identifying proteins that differentially interact with Srsf3 in response to PDGF-AA ligand stimulation are ongoing and will shed light on these mechanisms.

This study represents the first endogenous CLIP analysis of Srsf3 in the absence of protein tagging, and thus circumvents potential limitations with prior approaches in which assayed RBPs were overexpressed and/or fused to another protein. Our eCLIP analyses revealed several changes in Srsf3 transcript binding downstream of PDGF-AA stimulation, including increased Srsf3 binding to exons and loss of Srsf3 binding to canonical CA-rich motifs. A previous CLIP study using a stably integrated SRSF3-EGFP transgene in mouse P19 cells determined that SRSF3 binding was enriched in exons, particularly within 100 nucleotides of the 5' and, to a lesser extent, 3' splice sites (*Änkö et al., 2012*), consistent with our results. This same study identified a CA-rich canonical SRSF3 motif (*Änkö et al., 2012*). While such motifs were identified here in the absence of PDGF-AA treatment, they were lost

upon ligand stimulation. Again, this shift could be due to loss of RNA binding owing to electrostatic repulsion and/or changes in ribonucleoprotein composition and will be the subject of future studies.

Our findings additionally pointed to novel properties of exons whose inclusion is dependent on Srsf3 in the absence of PDGFRα signaling. We demonstrated that these included exons had a higher GC content, a lower ratio of downstream intron to exon GC content, and a decreased ratio of upstream and downstream intron to exon length. These findings are consistent with previous results demonstrating that included exons tend to have higher GC content than the flanking introns (*Amit et al., 2012*). Of note, PDGF-AA ligand stimulation resulted in binding of Srsf3 to exons with a lower GC content, again suggesting that phosphorylation of the RBP downstream of this signaling axis promotes exon skipping.

By cross-referencing our RNA-seq and eCLIP datasets, we showed that 24% of the alternatively spliced transcripts across our four treatment comparisons had an Srsf3 eCLIP peak. As Srsf3 also has functions in the cytoplasm, such as RNA trafficking, translation, and degradation (*Howard and Sanford, 2015*), the additional eCLIP peaks may reflect alternate roles for Srsf3 in RNA metabolism. Conversely, Srsf3-mediated AS may be delayed following transcript binding in the short timeframe of our experimental design. However, the extent of overlap that we observed is in line with previous studies correlating alternatively spliced transcripts upon knockdown of an RBP with endogenous eCLIP results for that same RBP, including Rbfox2 (10%) (*Moss et al., 2023*) and LUC7L2 (18–26%) (*Jourdain et al., 2021*). The degree to which our RNA-seq and eCLIP datasets overlapped here points to the robustness and biological significance of our findings.

Our combined analyses demonstrated that Srsf3 binds and mediates the AS of transcripts that encode proteins that regulate PI3K signaling and early endosomal trafficking downstream of PDGFRα activation, including *Becn1*, *Wdr81*, and *Acap3*. Homozygous missense mutations in *WDR81* cause cerebellar ataxia, impaired intellectual development, and dysequilibrium syndrome 2 (OMIM 610185) in humans (*Gulsuner et al., 2011*), with some patients exhibiting coarse facial features and strabismus (*Garcias and Roth, 2007*), pointing to a critical role for this gene product in craniofacial development. Consistently, we found that related GO terms, such as phosphatidylinositol phosphate binding and endosome to lysosome transport, were significantly enriched among alternatively spliced transcripts in murine embryonic facial mesenchyme upon loss of PI3K binding to PDGFRα and/or knockdown of *Srsf3* (*Dennison et al., 2021*). These data further confirm that our iMEPM model system served as a powerful platform to uncover mechanisms that are utilized during craniofacial development in vivo.

Our subsequent in vitro validation studies showed that Srsf3-mediated PDGFRα signaling results in enlarged early endosomes, retention of the receptor in these early endosomes, and increased downstream PI3K-mediated Akt signaling. Relatedly, we and others have linked spatial organization with the propagation of PDGFRα signaling, such that rapid internalization of the receptors into early endosomes or autophagy of the receptors are required for maximal phosphorylation of AKT downstream of PDGFRα activation (*Rogers et al., 2022*; *Simpson et al., 2024*). Together, these results indicate a feedback loop at play in the craniofacial mesenchyme in which stimulation of PDGFRα homodimer signaling leads to Srsf3-dependent AS of transcripts, a subsequent increase in the levels of proteins that maintain the receptor in early endosomes, and a corresponding decrease in the levels of proteins that promote trafficking of the receptor to late endosomes for eventual degradation. These findings thus represent a novel mechanism by which PDGFRα activity is maintained and propagated within the cell. Whether similar mechanisms exist downstream of alternate RTKs or contribute to increases in the phosphorylation of effector proteins other than Akt remain to be determined. In the future, it will be worthwhile to attempt to functionally link the AS of transcripts such as *Becn1*, *Wdr81*, and/or *Acap3* to the endosomal trafficking changes observed above using splice-switching antisense oligonucleotides (ASOs).

Taken together, our findings significantly enhance our understanding of the molecular mechanisms by which Srsf3 activity is regulated downstream of growth factor stimulation. Interestingly, a recent study demonstrated that retention of another shuttling SR protein, Srsf1, exclusively in the nucleus resulted in widespread ciliary defects in mice (*Haward et al., 2021*). This finding indicates that dissecting nuclear from cytoplasmic functions of SR proteins can provide powerful insight into the physiological relevance of each. Going forward, it will be critical to explore the in vivo consequences of abrogating Akt-mediated phosphorylation of Srsf3 and comparing the resulting phenotype to those of embryos with constitutive or conditional ablation of *Srsf3* in the neural crest lineage (*Dennison*

*et al., 2021; Jumaa et al., 1999*). These experiments are ongoing and should shed considerable light on the roles of RBP post-translational modifications during development.

## Materials and methods

### Generation of scramble and *Srsf3* shRNA iMEPM cell lines

iMEPM cells were derived from a male *Cdkn2a*[-/-] embryo as previously described (*Fantauzzo and Soriano, 2017*). iMEPM cells were cultured in growth medium (Dulbecco's modified Eagle's medium [Gibco, Thermo Fisher Scientific, Waltham, MA] supplemented with 50 U/mL penicillin [Gibco], 50 µg/mL streptomycin [Gibco], and 2 mM L-glutamine [Gibco] containing 10% fetal bovine serum [FBS] [Hyclone Laboratories Inc, Logan, UT]) and grown at 37°C in 5% carbon dioxide. iMEPM cells were tested for mycoplasma contamination using the MycoAlert Mycoplasma Detection Kit (Lonza Group Ltd, Basel, Switzerland) and confirmed negative for mycoplasma contamination. Packaged lentiviruses containing pLV[shRNA]-EGFP:T2A:Puro-U6>Scramble_shRNA (vectorID: VB010000-0009mxc) with sequence CCTAAGGTTAAGTCGCCCTCGCTCGAGCGAGGGCGACTTAACCTTAGG or pLV[shRNA]-EGFP:T2A:Puro-U6>mSrsf3[shRNA#1] (vectorID: VB900060-7699yyh) with sequence GAATGATAAAGCGGTGTTTACTCGAGTAAACACCGCTTTATCATTCC were purchased from VectorBuilder (Chicago, IL). Medium containing lentivirus for a multiplicity of infection of 10 for 200,000 cells with the addition of 10 ug/mL polybrene was added to iMEPM cells for 16 hr, and cells were subsequently grown in the presence of 4 ug/mL puromycin for 10 days. Cells with the highest GFP expression (top 20%) were isolated on a Moflo XDP 100 cell sorter (Beckman Coulter Inc, Brea, CA) and expanded. Srsf3 expression in scramble and *Srsf3* shRNA cell lines was confirmed by western blotting. Once the stable cell lines were established, they were split at a ratio of 1:4 for maintenance. Scramble and *Srsf3* shRNA cells were used for experiments at passages 9–20.

### Immunoprecipitation and western blotting

To induce PDGFRα signaling, cells at ~70% confluence were serum starved for 24 hr in starvation medium (Dulbecco's modified Eagle's medium [Gibco] supplemented with 50 U/mL penicillin [Gibco], 50 µg/mL streptomycin [Gibco], and 2 mM L-glutamine [Gibco] containing 0.1% FBS [Hyclone Laboratories Inc.]) and stimulated with 10 ng/mL rat PDGF-AA ligand (R&D Systems, Minneapolis, MN) diluted from a 1.5 µg/mL working solution in 40 nM HCl containing 0.1% BSA for the indicated length of time. When applicable, UV-crosslinking was performed at 254 nm and 400 mJ/cm$^2$ using a Vari-X-Link system (UVO3 Ltd, Cambridgeshire, UK). For immunoprecipitation of Srsf3, cells were resuspended in ice-cold CLIP lysis buffer (50 mM Tris-HCl pH 7.4, 100 mM NaCl, 1% NP-40, 0.1% SDS, 0.5% sodium deoxycholate, 1× cOmplete Mini protease inhibitor cocktail [Roche, MilliporeSigma, Burlington, MA], 1 mM PMSF, 10 mM NaF, 1 mM Na$_3$VO$_4$, 25 mM β-glycerophosphate). Cleared lysates were collected by centrifugation at 18,000 × *g* for 20 min at 4°C. Anti-Srsf3 antibody (10 µg/sample) (ab73891, Abcam, Waltham, MA) was added to protein A Dynabeads (125 µL/sample) (Invitrogen, Thermo Fisher Scientific) washed twice in ice-cold CLIP lysis buffer and incubated for 45 min at room temperature. Cell lysates were incubated with antibody-conjugated Dynabeads or Dynabeads M-280 sheep anti-rabbit IgG (Invitrogen, Thermo Fisher Scientific) washed twice in ice-cold CLIP lysis buffer overnight at 4°C. The following day, Dynabeads were washed twice each with ice-cold high salt wash buffer (50 mM Tris-HCl pH 7.4, 1 M NaCl, 1 mM EDTA, 1% NP-40, 0.1% SDS, 0.5% sodium deoxycholate) followed by ice-cold wash buffer (20 mM Tris-HCl pH 7.4, 10 mM MgCl$_2$, 0.2% Tween-20). The precipitated proteins were eluted with 1× NuPAGE LDS buffer (Thermo Fisher Scientific) containing 100 mM dithiothreitol, heated for 10 min at 70°C, and separated by SDS-PAGE. For western blotting of whole-cell lysates, protein lysates were generated by resuspending cells in ice-cold NP-40 lysis buffer (20 mM Tris-HCl pH 8, 150 mM NaCl, 10% glycerol, 1% NP-40, 2 mM EDTA, 1× cOmplete Mini protease inhibitor cocktail [Roche], 1 mM PMSF, 10 mM NaF, 1 mM Na$_3$VO$_4$, 25 mM β-glycerophosphate) and collecting cleared lysates by centrifugation at 13,400 × *g* for 20 min at 4°C. Laemmli buffer containing 10% β-mercaptoethanol was added to the lysates, which were heated for 5 min at 100°C. Proteins were subsequently separated by SDS-PAGE. Western blot analysis was performed according to standard protocols using horseradish peroxidase-conjugated secondary antibodies. Blots were imaged using a ChemiDoc XRS+ (Bio-Rad Laboratories, Inc, Hercules, CA) or a ChemiDoc (Bio-Rad Laboratories, Inc). The following primary

antibodies were used for western blotting: Srsf3 (1:1000, ab73891, Abcam), Gapdh (1:50,000, 60004-1-Ig, Proteintech Group, Inc, Rosemont, IL), Wdr81 (1:1000, 24874-1-AP, Proteintech Group, Inc), phospho-Akt (Ser473) (1:1000, 9271, Cell Signaling Technology, Inc, Danvers, MA), Akt (1:1000, 9272, Cell Signaling Technology, Inc), peroxidase AffiniPure goat anti-mouse IgG (H+L) (1:20,000, 115035003, Jackson ImmunoResearch Inc, West Grove, PA), peroxidase AffiniPure goat anti-rabbit IgG (H+L) (1:20,000, 111035003, Jackson ImmunoResearch Inc). Quantifications of signal intensity were performed with ImageJ software (version 1.53t, National Institutes of Health, Bethesda, MD). The relative ratio of Wdr81 isoforms was calculated as the fraction of the larger isoform divided by the fraction of the smaller isoform following normalization to Gapdh levels. Relative phospho-Akt levels were determined by normalizing to total Akt levels. When applicable, statistical analyses were performed with Prism 10 (GraphPad Software Inc, San Diego, CA) using a two-tailed, ratio paired *t*-test within each cell line and a two-tailed, unpaired *t*-test with Welch's correction between each cell line. Immunoprecipitation and western blotting experiments were performed across at least three independent experiments.

## RNA-sequencing and related bioinformatics analyses

$8 \times 10^5$ cells obtained from each of three independent biological replicates per treatment were frozen on liquid nitrogen and stored at –80°C. Following thawing, total RNA was simultaneously isolated from all samples using the RNeasy Mini Kit (QIAGEN, Inc, Germantown, MD) according to the manufacturer's instructions. RNA was forwarded to the University of Colorado Cancer Center Genomics Shared Resource for quality control, library preparation, and sequencing. RNA purity, quantity, and integrity were assessed with a NanoDrop (Thermo Fisher Scientific) and a 4200 TapeStation System (Agilent Technologies, Inc, Santa Clara, CA) prior to library preparation. Total RNA (200 ng) was used for input into the Universal Plus mRNA-Seq kit with NuQuant (Tecan Group Ltd., Männedorf, Switzerland). Dual index, stranded libraries were prepared and sequenced on a NovaSeq 6000 Sequencing System (Illumina, San Diego, CA) to an average depth of ~54 million read pairs (2 × 150 bp reads).

Raw sequencing reads were de-multiplexed using bcl2fastq (Illumina). Trimming, filtering, and adapter contamination removal were performed using BBDuk (from the BBmap v35.85 tool suite) (*Bushnell, 2015*). For differential expression analysis, transcript abundance was quantified using Salmon (v1.4.0) (*Patro et al., 2017*) and a decoy-aware transcriptome index prepared using GENCODE (*Frankish et al., 2019*) GRCm39 M26. Gene-level summaries were calculated using tximport (*Soneson et al., 2015*) in R and differential expression was measured using DESeq2 (v.1.32.0) (*Love et al., 2014*). Significant changes in gene-level expression are reported for cases with adjusted p≤0.05 and fold change |FC| ≥ 2. Spearman correlation was computed between conditions for differentially expressed genes. For alternative splicing analysis, raw FASTQ were trimmed to a uniform length of 125 bp. Reads were aligned to the mouse genome (GRCm39 Gencode M26) using STAR (v.2.7.9a) (*Dobin et al., 2013*). Additional parameters for STAR: `--outFilterType BySJout --outFilterMismatchNmax 10 --outFilterMismatchNover-Lmax 0.04 --alignEndsType EndToEnd --runThreadN 16 --alignSJDBoverhangMin 4 --alignIntronMax 300000 --alignSJoverhangMin 8 --alignIntronMin 20`. All splice junctions detected in at least one read from the first-pass alignment were used in a second-pass alignment, per software documentation. Alternative splicing events were detected using rMATS (v4.0.2, default parameters plus '—cstat 0.0001') (*Shen et al., 2014*). Reads mapping to the splice junction as well as those mapping to the exon body were used in downstream analyses. Detected events were compared between treatment groups and considered significant with false discovery rate (FDR) ≤ 0.05, a difference in percent spliced in (|ΔPSI|) ≥ 0.05, and event detection in at least 10 reads in either condition. Raw read pairs, trimmed read pairs for Salmon input, Salmon mapping rate per sample, trimmed read pairs (125 bp) for STAR input, and STAR unique mapping rate are indicated below. Gene ontology analysis was performed with various libraries from the Enrichr gene list enrichment analysis tool (*Chen et al., 2013*; *Kuleshov et al., 2016*) and terms with p<0.05 were considered significant.

| Sample | Raw read pairs | Trimmed read pairs for Salmon input | Salmon mapping rate | Trimmed read pairs (125 bp) for STAR input | STAR unique mapping rate |
|---|---|---|---|---|---|
| -PDGF-AA scramble_1 | 47181591 | 44410442 | 0.89055 | 36343779 | 0.8773 |
| -PDGF-AA scramble_2 | 54612492 | 50500367 | 0.878847 | 39971864 | 0.8681 |
| -PDGF-AA scramble_3 | 69353787 | 65529075 | 0.912399 | 48327896 | 0.9022 |
| -PDGF-AA shSrsf3_1 | 91657568 | 84086217 | 0.913324 | 61269254 | 0.9035 |
| -PDGF-AA shSrsf3_2 | 77309551 | 71220292 | 0.91638 | 49390634 | 0.9013 |
| -PDGF-AA shSrsf3_3 | 42645900 | 41078549 | 0.910338 | 28737018 | 0.9054 |
| +PDGF-AA scramble_1 | 71080979 | 66836059 | 0.916828 | 48116755 | 0.9027 |
| +PDGF-AA scramble_2 | 69667521 | 64974624 | 0.890505 | 47762451 | 0.884 |
| +PDGF-AA scramble_3 | 78680108 | 72721916 | 0.911689 | 52936280 | 0.9008 |
| +PDGF-AA shSrsf3_1 | 42776470 | 41165756 | 0.914797 | 28076373 | 0.9019 |
| +PDGF-AA shSrsf3_2 | 37944773 | 35759828 | 0.908637 | 23528077 | 0.8987 |
| +PDGF-AA shSrsf3_3 | 36391090 | 34257983 | 0.911463 | 26455873 | 0.8995 |

## qPCR

Total RNA was isolated using the RNeasy mini kit (QIAGEN) according to the manufacturer's instructions. First-strand cDNA was synthesized using a ratio of 2:1 random primers:oligo (dT)$_{12-18}$ primer and SuperScript II RT (Invitrogen, Thermo Fisher Scientific) according to the manufacturer's instructions. All reactions were performed with 1× ThermoPol buffer [0.02 M Tris pH 8.8, 0.01 M KCl, 0.01 M (NH$_4$)$_2$SO$_4$, 2 mM MgSO$_4$, and 0.1% Triton X-100], 200 μM dNTPs, 200 nM primers (Integrated DNA Technologies, Inc, Coralville, IA), 0.6 U Taq polymerase, and 1 μg cDNA in a 25 μL reaction volume. The primers used are indicated below. The following PCR protocol was used for *Arhgap12*: step 1, 3 min at 94°C; step 2, 30 s at 94°C; step 3, 30 s at 47°C; step 4, 30 s at 72°C; repeat steps 2–4 for 34 cycles; and step 5, 5 min at 72°C. The following PCR protocol was used for *Cep55*: step 1, 3 min at 94°C; step 2, 30 s at 94°C; step 3, 30 s at 48°C; step 4, 30 s at 72°C; repeat steps 2–4 for 34 cycles; and step 5, 5 min at 72°C. Two-thirds of total PCR products were electrophoresed on a 2% agarose/TBE gel containing ethidium bromide and photographed on an Aplegen Omega Fluor Gel Documentation System (Aplegen Inc, Pleasanton, CA). Quantifications of band intensity were performed with ImageJ software (version 1.53t, National Institutes of Health). The following PCR protocol was used for *Wdr81*: step 1, 3 min at 94°C; step 2, 30 s at 94°C; step 3, 30 s at 50°C; step 4, 30 s at 72°C; repeat steps 2–4 for 24 cycles; and step 5, 2 min at 72°C. PCR products were purified with AMPure XP Reagent (Beckman Coulter) and analyzed on a 4150 TapeStation System (Agilent Technologies, Inc) using High Sensitivity D1000 Screen-Tape (Agilent Technologies, Inc). The PSI was calculated independently for each sample as the percentage of the larger isoform divided by the total abundance of all isoforms within the given gel lane or TapeStation sample. Statistical analyses were performed with Prism 10 (GraphPad Software) using a two-tailed, unpaired *t*-test with Welch's correction. qPCR reactions were performed using three biological replicates.

| Transcript | Forward primer (5' to 3') | Reverse primer (5' to 3') |
| --- | --- | --- |
| *Arhgap12* | GGAGACATAGCACCATTGTG | GCACTGCCCAAGAAGACAAC |
| *Cep55* | CCTTTCGGCTCCTTTGAACT | GCAGTGTCTGACTTGGAGCT |
| *Wdr81* | GCTTTGTGGACTGCAGGAAG | GCAGGGAACAGACACCAATC |

## Enhanced UV-crosslinking and immunoprecipitation and related bioinformatics analyses

Experiments were performed as previously described in biological duplicates (*Van Nostrand et al., 2017*; *Van Nostrand et al., 2016*). Briefly, 2 million cells per treatment were serum starved and treated with 10 ng/mL PDGF-AA as described above. Cells were subsequently UV-crosslinked at 254 nm and 400 mJ/cm$^2$, scraped in 1× phosphate-buffered saline (PBS), and transferred to 1.5 mL Eppendorf tubes, at which point excess PBS was removed and cells were frozen on liquid nitrogen and stored at –80°C. Following thawing, cells were lysed in ice-cold CLIP lysis buffer, sonicated by BioRuptor (Diagenode, Denville, NJ) and treated with RNase I (Invitrogen, Thermo Fisher Scientific). Two percent of lysates were set aside as size-matched input samples. Srsf3-RNA complexes were immunoprecipitated with anti-Srsf3 antibody (10 µg per sample) (ab73891, Abcam) conjugated to protein A Dynabeads (Invitrogen, Thermo Fisher Scientific). IP samples were washed and dephosphorylated with FastAP (Thermo Fisher Scientific) and T4 PNK (New England Biolabs, Ipswich, MA). IP samples underwent on-bead ligation of barcoded RNA adapters (/5phos/rArGrArUrCrGrGrArArGrArGrCrGrUrCrGrUrG/3SpC3/) to the 3' end using T4 RNA ligase (New England Biolabs). Following elution, protein-RNA complexes were run on 4–12% Bis-Tris 1.5 mm gels (Thermo Fisher Scientific) and transferred onto nitrocellulose membranes. The 20–75 kDa region was excised and digested with proteinase K (New England Biolabs). RNA was isolated with acid phenol/chloroform/isoamyl alcohol (pH 6.5) (Thermo Fisher Scientific), reverse transcribed with Superscript IV RT (Invitrogen, Thermo Fisher Scientific), and treated with ExoSAP-IT (Applied Biosystems, Thermo Fisher Scientific) to remove excess primers and unincorporated nucleotides. Samples underwent 3' ligation of barcoded DNA adapters (/5Phos/NNNNNNNNNNAGATCGGAAGAGCACACGTCTG/3SpC3/), clean-up with Dynabeads MyOne Silane (Invitrogen, Thermo Fisher Scientific), and qPCR to determine the appropriate number of PCR cycles. Libraries were then amplified with Q5 PCR mix (New England Biolabs) for a total of 16–25 cycles. Libraries were forwarded to the University of Colorado Cancer Center Genomics Shared Resource for quality control and sequencing. Sample integrity was assessed with a D1000 ScreenTape System (Agilent Technologies, Inc) and sequenced on a NovaSeq 6000 Sequencing System (Illumina) to an average depth of ~20 million read pairs (2 × 150 bp reads).

Raw sequencing reads were de-multiplexed using bcl2fastq (Illumina). Adapters were trimmed using cutadapt (v.1.18) (*Martin, 2011*). Trimmed reads were quality filtered and collapsed using a combination of FASTX-Toolkit (RRID:SCR_005534, v.0.0.14, http://hannonlab.cshl.edu/fastx_toolkit), seqtk (RRID:SCR_018927, v.1.3-r106, https://github.com/lh3/seqtk; *Li, 2018*), and custom scripts. After collapsing the reads, unique molecular identifiers were removed using seqtk. STAR index for repetitive elements was created using repetitive sequences from msRepDB (*Liao et al., 2022*). Reads ≥18 nt were mapped to the repetitive elements using STAR (v.2.7.9a) (*Dobin et al., 2013*). Reads unmapped to the repetitive elements were mapped to the mouse genome (GRCm39 Gencode M26) using STAR (v.2.7.9a) with parameters alignEndsType: EndtoEnd and outFilterMismatchNoverReadLmax: 0.04. Peaks were called using omniCLIP (v.0.20) (*Drewe-Boss et al., 2018*) with the foreground penalty (--fg_pen) parameter set to 5. Peaks were annotated and motif analyses performed using RCAS (v.1.19.0) (*Uyar et al., 2017*) and custom R script. Motif enrichment significance was calculated using a *t*-test. For visualization purposes, bigWig files were created from bam files using deepTools (v.3.5.5) (*Ramírez et al., 2016*). Peaks were visualized in Integrative Genomics Viewer (v.2.13.0) (*Robinson et al., 2011*). Intron and exon features were calculated using Matt (v.1.3.1) (*Gohr and Irimia, 2019*), and statistical analyses were performed using a Mann–Whitney *U*-test. For overlap of eCLIP peaks and alternative splicing events, peak coordinates were taken from omniCLIP bed files and alternative splicing coordinates were taken from rMATS output. Overlapping coordinates from alternative splicing events were defined following the rMAPS default values (*Park et al., 2016*). Overlap was calculated using valr (v.0.6.4) (*Riemondy et al., 2017*) and custom R scripts. Raw read pairs, trimmed read pairs, collapsed reads, reads after removing repetitive elements, mapped reads,

peaks, and annotated peaks are indicated below. Gene ontology analysis was performed with various libraries from the Enrichr gene list enrichment analysis tool (*Chen et al., 2013*; *Kuleshov et al., 2016*) and terms with p<0.05 were considered significant.

| Sample | Raw read pairs | Trimmed read pairs | Collapsed reads | Reads after removing repetitive elements | Mapped reads | Peaks | Annotated peaks |
|---|---|---|---|---|---|---|---|
| -PDGF-AA size-matched input | 34303575 | 22904092 | 13449745 | 13358235 | 17206 | | |
| -PDGF-AA replicate 1 | 22983544 | 18369023 | 2758371 | 2758371 | 440436 | | |
| -PDGF-AA replicate 2 | 15666256 | 12263540 | 2742638 | 2065674 | 388996 | 6969 | 6607 |
| +PDGF-AA size-matched input | 52420337 | 37454316 | 13811675 | 13643948 | 24275 | | |
| +PDGF-AA | 30105052 | 23355466 | 3417325 | 2845801 | 872085 | 9075 | 8623 |

## Immunofluorescence analysis

Cells were seeded onto glass coverslips at ~40% confluence per 24-well plate well in iMEPM growth medium. After 24 hr, cells were serum starved and treated with 10 ng/mL PDGF-AA as described above. Cells were fixed in 4% paraformaldehyde (PFA) in PBS with 0.1% Triton X-100 for 10 min and washed in PBS. Cells were blocked for 1 hr in 5% normal donkey serum (Jackson ImmunoResearch Inc) in PBS and incubated overnight at 4°C in primary antibody diluted in 1% normal donkey serum in PBS. After washing in PBS, cells were incubated in donkey anti-rabbit IgG (H+L) highly cross-absorbed secondary antibody, Alexa Fluor 488 (1:1000; A21206; Invitrogen) or donkey anti-mouse IgG (H+L) highly cross-absorbed secondary antibody, Alexa Fluor 546 (1:1000; A10036; Invitrogen) diluted in 1% normal donkey serum in PBS with 2 µg/mL DAPI (Sigma-Aldrich, St. Louis, MO) for 1 hr. Cells were mounted in VECTASHIELD HardSet Antifade Mounting Medium (Vector Laboratories, Inc, Burlingame, CA) and photographed using an Axiocam 506 mono digital camera (Carl Zeiss Microscopy LLC, White Plains, NY) fitted onto an Axio Observer 7 fluorescence microscope (Carl Zeiss Microscopy LLC) with the ×63 oil objective with a numerical aperture of 1.4 at room temperature. The following antibodies were used for immunofluorescence analysis: Rab5 (1:200, C8B1, 3547, Cell Signaling Technology Inc), PDGFRα (1:20, AF1062, R&D Systems). For assessment of Rab5 puncta size and colocalization experiments, three independent trials, or biological replicates, were performed. For each biological replicate, 20 technical replicates consisting of individual cells were imaged with Z-stacks (0.24 µm between Z-stacks with a range of 1–6 Z-stacks) per timepoint. Images were deconvoluted using ZEN Blue software (Carl Zeiss Microscopy LLC) using the 'Better, fast (Regularized Inverse Filter)' setting. Extended depth of focus was applied to Z-stacks using ZEN Blue software (Carl Zeiss Microscopy LLC) to generate images with the maximum depth of field. For assessment of Rab5 puncta size, images were converted to 8-bit using Fiji software (version 2.14.0/1.54f). Images were subsequently converted to a mask and watershed separation was applied. A region of interest (ROI) was drawn around each Rab5-positive cell and particles were analyzed per cell using the 'analyze particles' function. For colocalization measurements, an ROI was drawn around each PDGFRα-positive cell in the corresponding Cy3 channel using Fiji. For each image with a given ROI, the Cy3 channel and the EGFP channel were converted to 8-bit images. Colocalization was measured using the Colocalization Threshold function, where the rcoloc value (PCC) was used in statistical analysis. Statistical analyses were performed on the average values from each biological replicate with Prism 10 (GraphPad Software Inc) using a two-way ANOVA followed by uncorrected Fisher's LSD test.

## Acknowledgements

We are grateful to Jessica Johnston and Erin Binne for technical assistance, and Drs. Allison Swain, Justin Roberts, and Aaron Johnson at the University of Colorado Anschutz Medical Campus for advice on eCLIP experiments. Cell sorting was performed at the University of Colorado Cancer Center Flow Cytometry Shared Resource with assistance from Dr. Dmitry Baturin. RNA-seq and eCLIP sequencing experiments were performed at the University of Colorado Cancer Center Genomics Shared Resource. We thank members of the Fantauzzo laboratory for their critical comments on the manuscript. This work was supported by the National Institutes of Health grants R01DE030864 (to KAF), R35GM147025 (to NM), the University of Colorado Anschutz Medical Campus RNA Bioscience

Initiative (to NM and MPS), and F31DE032252 (to TEF). The Flow Cytometry Shared Resource and Genomics Shared Resource are supported by the National Institutes of Health grant P30CA046934.

## Additional information

### Funding

| Funder | Grant reference number | Author |
| --- | --- | --- |
| National Institutes of Health | R01DE030864 | Katherine A Fantauzzo |
| National Institutes of Health | R35GM147025 | Neelanjan Mukherjee |
| University of Colorado Anschutz Medical Campus | | Marcin P Sajek Neelanjan Mukherjee |
| National Institutes of Health | F31DE032252 | Thomas E Forman |

The funders had no role in study design, data collection and interpretation, or the decision to submit the work for publication.

### Author contributions

Thomas E Forman, Conceptualization, Formal analysis, Funding acquisition, Investigation, Visualization, Methodology, Writing – original draft; Marcin P Sajek, Conceptualization, Formal analysis, Investigation, Visualization, Methodology, Writing – review and editing; Eric D Larson, Formal analysis, Investigation, Writing – review and editing; Neelanjan Mukherjee, Conceptualization, Supervision, Methodology, Writing – review and editing; Katherine A Fantauzzo, Conceptualization, Formal analysis, Supervision, Funding acquisition, Visualization, Methodology, Writing – original draft, Project administration

### Author ORCIDs

Thomas E Forman ⓘ https://orcid.org/0000-0002-8912-3503
Marcin P Sajek ⓘ https://orcid.org/0000-0002-4115-4191
Eric D Larson ⓘ https://orcid.org/0000-0002-2881-4861
Neelanjan Mukherjee ⓘ https://orcid.org/0000-0003-0017-1400
Katherine A Fantauzzo ⓘ https://orcid.org/0000-0003-0365-4168

Reviewer #1 (Public review): https://doi.org/10.7554/eLife.98531.3.sa1
Reviewer #2 (Public review): https://doi.org/10.7554/eLife.98531.3.sa2
Author response https://doi.org/10.7554/eLife.98531.3.sa3

## Additional files

### Supplementary files
• MDAR checklist

### Data availability

All data generated or analyzed during this study are included in the manuscript and supporting files; source data files have been provided for *Figures 1–6*. The eCLIP and RNA-sequencing datasets generated during this study have been deposited in GEO under SuperSeries accession number GSE263170. Custom analysis scripts are available through GitHub, copy archived at *Sajek and Larson, 2024*. Scramble iMEPM and *Srsf3* shRNA iMEPM cell lines are available upon request from Dr. Fantauzzo.

The following dataset was generated:

| Author(s) | Year | Dataset title | Dataset URL | Database and Identifier |
|---|---|---|---|---|
| Forman TE, Sajek MP, Larson ED, Mukherjee N, Fantauzzo KA | 2024 | PDGFRa signaling regulates Srsf3 transcript binding to affect PI3K signaling and endosomal trafficking | https://www.ncbi.nlm.nih.gov/geo/query/acc.cgi?acc=GSE263170 | NCBI Gene Expression Omnibus, GSE263170 |

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

# Appendix 1

## Appendix 1—key resources table

| Reagent type (species) or resource | Designation | Source or reference | Identifiers | Additional information |
|---|---|---|---|---|
| Cell line (*Mus musculus*, male) | iMEPM | *Fantauzzo and Soriano, 2017* | | |
| Cell line (*M. musculus*) | Scramble iMEPM | This paper | | Cell line maintained in K. Fantauzzo lab |
| Cell line (*M. musculus*) | *Srsf3* shRNA iMEPM | This paper | | Cell line maintained in K. Fantauzzo lab |
| Antibody | Srsf3 (rabbit polyclonal) | Abcam | Abcam:ab73891 | IP (10 µg/sample), WB (1:1000) |
| Antibody | Gapdh (mouse monoclonal) | Proteintech Group, Inc | Proteintech:60004-1-Ig | WB (1:50,000) |
| Antibody | Wdr81 (rabbit polyclonal) | Proteintech Group, Inc | Proteintech:24874-1-AP | WB (1:1000) |
| Antibody | Phospho-Akt (rabbit polyclonal) | Cell Signaling Technology | Cell Signaling Technology:9271S | WB (1:1000) |
| Antibody | Akt (rabbit polyclonal) | Cell Signaling Technology | Cell Signaling Technology:9272S | WB (1:1000) |
| Antibody | Peroxidase AffiniPure Goat Anti-Mouse IgG (H+L) (goat polyclonal) | Jackson ImmunoResearch Inc | Jackson ImmunoResearch:115-035-003 | WB (1:20,000) |
| Antibody | Peroxidase AffiniPure Goat Anti-Rabbit IgG (H+L) (goat polyclonal) | Jackson ImmunoResearch Inc | Jackson ImmunoResearch:111-035-003 | WB (1:20,000) |
| Antibody | Rab5 (rabbit monoclonal) | Cell Signaling Technology | Cell Signaling Technology:3547S | IF (1:200) |
| Antibody | PDGFRα (mouse polyclonal) | R&D Systems | R&D Systems:AF1062 | IF (1:20) |
| Antibody | Donkey anti-Rabbit IgG (H+L) Highly Cross-Absorbed Secondary Antibody, Alexa Fluor 488 (donkey polyclonal) | Invitrogen | Invitrogen:A21206 | IF (1:1000) |
| Antibody | Donkey anti-Mouse IgG (H+L) Highly Cross-Absorbed Secondary Antibody, Alexa Fluor 546 (donkey polyclonal) | Invitrogen | Invitrogen:A10036 | IF (1:1000) |
| Recombinant DNA reagent | pLV[shRNA]-EGFP:T2A:Puro-U6>Scramble_shRNA | VectorBuilder | Vector Builder:VB010000-0009mxc | CCTAAGGTTAAGTCGCCCTCGCTCGAGCGAGGGCGACTTAACCTTAGG |
| Recombinant DNA reagent | pLV[shRNA]-EGFP:T2A:Puro-U6>mSrsf3[shRNA#1] | VectorBuilder | Vector Builder:VB90060-7699yyh | GAATGATAAAGCGGTGTTTACTCGAGTAAACACCGCTTTATCATTCC |
| Sequence-based reagent | Random primers | Invitrogen | Invitrogen:48190011 | |
| Sequence-based reagent | oligo (dT)$_{12-18}$ primer | Invitrogen | Invitrogen:18418012 | |
| Sequence-based reagent | *Arhgap12*_F | This paper | PCR primer | GGAGACATAGCACCATTGTG |
| Sequence-based reagent | *Arhgap12*_R | This paper | PCR primer | GCACTGCCCAAGAAGACAAC |
| Sequence-based reagent | *Cep55*_F | This paper | PCR primer | CCTTTCGGCTCCTTTGAACT |
| Sequence-based reagent | *Cep55*_R | This paper | PCR primer | GCAGTGTCTGACTTGGAGCT |
| Sequence-based reagent | *Wdr81*_F | This paper | PCR primer | GCTTTGTGGACTGCAGGAAG |
| Sequence-based reagent | *Wdr81*_R | This paper | PCR primer | GCAGGGAACAGACACCAAT |

*Appendix 1 Continued on next page*

*Appendix 1 Continued*

| Reagent type (species) or resource | Designation | Source or reference | Identifiers | Additional information |
|---|---|---|---|---|
| Sequence-based reagent | RiL19 | This paper | Barcoded RNA adapters | /5phos/rArGrArUrCrGrGrArArGrArGrCrGrUrCrGrUrG/3SpC3/ |
| Sequence-based reagent | Rand103Tr3 | This paper | Barcoded DNA adapters | /5Phos/NNNNNNNNNNAGATCGGAAGAGCACACGTCTG/3SpC3/ |
| Peptide, recombinant protein | PDGF-AA | R&D Systems | R&D:1055AA050 | 10 ng/mL |
| Peptide, recombinant protein | Protein A Dynabeads | Invitrogen | Invitrogen:10002D | IP (125 µL/sample) |
| Peptide, recombinant protein | Dynabeads M-280 sheep anti-rabbit IgG | Invitrogen | Invitrogen:11203D | IP (125 µL/sample) |
| Peptide, recombinant protein | SuperScript II RT | Invitrogen | Invitrogen:18064014 | |
| Peptide, recombinant protein | RNase I | Invitrogen | Invitrogen:AM2294 | |
| Peptide, recombinant protein | FastAP | Thermo Fisher Scientific | Thermo Fisher Scientific:EF0654 | |
| Peptide, recombinant protein | T4 PNK | New England Biolabs | New England Biolabs:M0201S | |
| Peptide, recombinant protein | T4 RNA ligase | New England Biolabs | New England Biolabs:M0437M | |
| Peptide, recombinant protein | Proteinase K | New England Biolabs | New England Biolabs:P8107S | |
| Peptide, recombinant protein | SuperScript IV | Invitrogen | Invitrogen:18090010 | |
| Chemical compound, drug | DAPI | Sigma-Aldrich | Sigma-Aldrich:D9542-10MG | 2 µg/mL |
| Commercial assay or kit | MycoAlert Mycoplasma Detection Kit | Lonza Group Ltd | Lonza:LT07-218 | |
| Commercial assay or kit | RNeasy Mini Kit | QIAGEN, Inc | QIAGEN:74104 | |
| Commercial assay or kit | Universal Plus mRNA-Seq kit with NuQuant | Tecan Group Ltd | Tecan Group:0361-A01 | |
| Commercial assay or kit | AMPure XP Reagent | Beckman Coulter | Beckman Coulter:A63880 | |
| Commercial assay or kit | ExoSAP-IT | Applied Biosystems | Applied Biosystems:78200.200.UL | |
| Commercial assay or kit | Dynabeads MyOne Silane | Invitrogen | Invitrogen:37002D | |
| Commercial assay or kit | Q5 PCR mix | New England Biolabs | New England Biolabs:M0492S | |
| Software, algorithm | ImageJ | NIH | RRID:SCR_003070 | |
| Software, algorithm | Prism 10 | GraphPad Software Inc. | RRID:SCR_000306 | |
| Software, algorithm | bcl2fastq | Illumina | RRID:SCR_015058 | |
| Software, algorithm | BBDuk | https://sourceforge.net/projects/bbmap/ | | |

*Appendix 1 Continued on next page*

*Appendix 1 Continued*

| Reagent type (species) or resource | Designation | Source or reference | Identifiers | Additional information |
|---|---|---|---|---|
| Software, algorithm | Salmon | *Patro et al., 2017* | | |
| Software, algorithm | GENCODE | *Frankish et al., 2019* | | |
| Software, algorithm | tximport | *Soneson et al., 2015* | | |
| Software, algorithm | DESeq2 | *Love et al., 2014* | | |
| Software, algorithm | STAR | *Dobin et al., 2013* | | |
| Software, algorithm | rMATS | *Shen et al., 2014* | | |
| Software, algorithm | Enrichr | *Chen et al., 2013*; *Kuleshov et al., 2016* | | |
| Software, algorithm | cutadapt | *Martin, 2011* | | |
| Software, algorithm | FASTX-Toolkit | http://hannonlab.cshl.edu/fastx_toolkit | | |
| Software, algorithm | seqtk | https://github.com/lh3/seqtk; *Li, 2018* | | |
| Software, algorithm | msRepDB | *Liao et al., 2022* | | |
| Software, algorithm | omniCLIP | *Drewe-Boss et al., 2018* | | |
| Software, algorithm | RCAS | *Uyar et al., 2017* | | |
| Software, algorithm | deepTools | *Ramírez et al., 2016* | | |
| Software, algorithm | Integrative Genomics Viewer | *Robinson et al., 2011* | | |
| Software, algorithm | Matt | *Gohr and Irimia, 2019* | | |
| Software, algorithm | rMAPS | *Park et al., 2016* | | |
| Software, algorithm | valr | *Riemondy et al., 2017* | | |
| Software, algorithm | ZEN Blue | Carl Zeiss Microscopy LLC | RRID:SCR_013672 | |
| Other | Normal donkey serum | Jackson ImmunoResearch Inc | Jackson ImmunoResearch:017-000-121 | |
| Other | VECTASHIELD HardSet Antifade Mounting Medium | Vector Laboratories, Inc | Vector Laboratories:H-1400–10 | |

