## [Editor Report · eLife Assessment]

This **fundamental** work provides new mechanistic insight in regulation of PDGF signaling through splicing controls. The evidence is **compelling** to demonstrate functional involvement of Srsf3, an RNA-binding protein, to this new and interesting mechanism. The work will be of broad interest to developmental biologists in general and molecular biologists/biochemists in the field of growth factor signaling and RNA processing.

---

## [Referee Report · Reviewer #1 (Public review)]

In their manuscript "PDGFRRa signaling regulates Srsf3 transcript binding to affect PI3K signaling and endosomal trafficking" Forman and colleagues use iMEPM cells to characterize the effects of PDGF signaling on alternative splicing. They first perform RNA-seq using a one-hour stimulation with Pdgf-AA in control and Srsf3 knockdown cells. While Srsf3 manipulation results in a sizeable number of DE genes, PDGF does not. They then turn to examine alternative splicing, due to findings from this lab. They find that both PDGF and Srsf3 contribute much more to splicing than transcription. They find that the vast majority of PDGF-mediated alternative splicing depends upon Srsf3 activity and that skipped exons are the most common events with PDGF stimulation typically promoting exon skipping in the presence of Srsf3. They used eCLIP to identify RNA regions bound to Srsf3. Under both PDGF conditions, the majority of peaks were in exons with +PDGF having a substantially greater number of these peaks. Interestingly, they find differential enrichment of sequence motifs and GC content in stimulated versus unstimulated cells. They examine 2 transcripts encoding PI3K pathway (enriched in their GO analysis) members: Becn1 and Wdr81. They then go on to examine PDGFRRa and Rab5, an endosomal marker, colocalization. They propose a model in which Srsf3 functions downstream of PDGFRRa signaling to, in part, regulate PDGFRa trafficking to the endosome. The findings are novel and shed light on the mechanisms of PDGF signaling and will be broadly of interest. This lab previously identified the importance of PDGF naling on alternative splicing. The combination of RNA-seq and eCLIP is an exceptional way to comprehensively analyze this effect. The results will be of great utility to those studying PDGF signaling or neural crest biology.

Comments on the revised version:

The authors have fully addressed my previous comments and I have no further concerns.

---

## [Referee Report · Reviewer #2 (Public review)]

Summary:

This manuscript builds upon the work of a previous study published by the group (Dennison, 2021) to further elucidate the coregulatory axis of Srsf3 and PDGFRa on craniofacial development. The authors in this study investigated the molecular mechanisms by which PDGFRa signaling activates the RNA-binding protein Srsf3 to regulate alternative splicing (AS) and gene expression (GE) necessary for craniofacial development. PDGFRa signaling-mediated Srsf3 phosphorylation drives its translocation into the nucleus and affect binding affinity to different proteins and RNA, but the exact molecular mechanisms were not known. The authors performed RNA sequencing on immortalized mouse embryonic mesenchyme (MEPM) cells treated with shRNA targeting 3' UTR of Srsf3 or scramble shRNA (to probe AS and DE events that are Srsf3 dependent) and with and without PDGF-AA ligand treatment (to probe AS and DE events that are PDGFRa signaling dependent). They found that PDGFRa signaling has more effect on AS than on DE. A matching eCLIP-seq experiment was performed to investigate how Srsf3 binding sites change with and without PDGFRa signaling.

Strengths:

(1) The work builds well upon the previous data and the authors employ a variety of appropriate techniques to answer their research questions.

(2) The authors show that Srsf3 binding pattern within the transcript as well as binding motifs change significantly upon PDGFRa signaling, providing a mechanistic explanation for the significant changes in AS.

(3) By combining RNA-seq and eCLIP datasets together, the authors identified a list of genes that are directly bound by Srsf3 and undergo changes in GE and/or AS. Two examples are Becn1 and Wdr81, which are involved in early endosomal trafficking.

Weaknesses:

(1) The authors identify two genes whose AS are directly regulated by Srsf3 and involved in endosomal trafficking; however, they do not validate the differential AS results and whether changes in these genes can affect endosomal trafficking. In Figure 6, they show that PDGFRa signaling is involved in endosome size and Rab5 colocalization, but do not show how Srsf3 and the two genes are involved.

(2) The proposed model does not account for other proteins mediating the activation of Srsf3 after Akt phosphorylation. How do we know this is a direct effect (and not secondary or tertiary effect)?

This is a thoroughly revised manuscript. I would like to congratulate the authors to have invested a lot of time, resources, new data, and a more refined discussion to make this a compelling piece of work. I have no further concerns.

---

## [Author Response]

The following is the authors’ response to the original reviews.

**Public Reviews:**

**Reviewer #1 (Public Review):**
In their manuscript "PDGFRRa signaling regulates Srsf3 transcript binding to affect PI3K signaling and endosomal trafficking" Forman and colleagues use iMEPM cells to characterize the effects of PDGF signaling on alternative splicing. They first perform RNA-seq using a one-hour stimulation with Pdgf-AA in control and Srsf3 knockdown cells. While Srsf3 manipulation results in a sizeable number of DE genes, PDGF does not. They then turn to examine alternative splicing, due to findings from this lab. They find that both PDGF and Srsf3 contribute much more to splicing than transcription. They find that the vast majority of PDGF-mediated alternative splicing depends upon Srsf3 activity and that skipped exons are the most common events with PDGF stimulation typically promoting exon skipping in the presence of Srsf3. They used eCLIP to identify RNA regions bound to Srsf3. Under both PDGF conditions, the majority of peaks were in exons with +PDGF having a substantially greater number of these peaks. Interestingly, they find differential enrichment of sequence motifs and GC content in stimulated versus unstimulated cells. They examine 2 transcripts encoding PI3K pathway (enriched in their GO analysis) members: Becn1 and Wdr81. They then go on to examine PDGFRRa and Rab5, an endosomal marker, colocalization. They propose a model in which Srsf3 functions downstream of PDGFRRa signaling to, in part, regulate PDGFRa trafficking to the endosome. The findings are novel and shed light on the mechanisms of PDGF signaling and will be broadly of interest. This lab previously identified the importance of PDGF naling on alternative splicing. The combination of RNA-seq and eCLIP is an exceptional way to comprehensively analyze this effect. The results will be of great utility to those studying PDGF signaling or neural crest biology. There are some concerns that should be considered, however.

We thank the Reviewer for these supportive comments.

(1) It took some time to make sense of the number of DE genes across the results section and Figure 1. The authors give the total number of DE genes across Srsf3 control and loss conditions as 1,629 with 1,042 of them overlapping across Pdgf treatment. If the authors would add verbiage to the point that this leaves 1,108 unique genes in the dataset, then the numbers in Figure 1D would instantly make sense. The same applies to PDGF in Figure 1F and the Venn diagrams in Figure 2.

We have edited the relevant sentence for Figure 1D as follows: “There was extensive overlap (521 out of 1,108; 47.0%) of Srsf3-dependent DE genes across ligand treatment conditions, resulting in a total of 1,108 unique genes within both datasets (Fig. 1C,D; Fig. S1A).” Similarly, we edited the relevant sentence for Figure 1F as follows: “There was limited overlap (4 out of 47; 8.51%) of PDGF-AA-dependent DE genes across Srsf3 conditions, resulting in a total of 47 unique genes within both datasets (Fig. 1E,F; Fig. S1B).” We edited the relevant sentence for Figure 2B as follows: “There was limited overlap (203 out of 1,705; 11.9%) of Srsf3-dependent alternatively-spliced transcripts across ligand treatment conditions, resulting in a total of 1,705 unique events within both datasets (Fig. 2A,B).” Finally, we edited the relevant sentence for Figure 2D as follows: “There was negligible overlap (9 out of 622; 1.45%) of PDGF-AA-dependent alternatively-spliced transcripts across Srsf3 conditions, resulting in a total of 622 unique events within both datasets (Fig. 2C,D).”

(2) The percentage of skipped exons in the +DPSI on the righthand side of Figure 2F is not readable.

We have moved the label for the percentage of skipped exon events with a +DPSI for the -PDGF-AA vs +PDGF-AA (scramble) alternatively-spliced transcripts in Figure 2E so that it is legible.

(3) It would be useful to have more information regarding the motif enrichment in Figure 3. What is the extent of enrichment? The authors should also provide a more complete list of enriched motifs, perhaps as a supplement.

We have added *P* values beneath the motifs in Figure 3F and 3G. Further, we have added a new Supplementary Figure, Figure S5, that lists the occurrence of the top 10 most enriched motifs in the unstimulated and, separately, stimulated samples in the eCLIP dataset and in a control dataset, as well as their *P* values.

(4) It is unclear what subset of transcripts represent the "overlapping datasets" on lines 280-315. The authors state that there are 149 unique overlapping transcripts, but the Venn diagram shows 270. Also, it seems that the most interesting transcripts are the 233 that show alternative splicing and are bound by Srsf3. Would the results shown in Figure 5 change if the authors focused on these transcripts?

The Reviewer is correct that 233 of the alternatively-spliced transcripts had an Srsf3 eCLIP peak, as indicated in Figure 5A. However, several of these eCLIP peaks were a large distance from an alternatively-spliced element in the rMATS datasets, indicating that Srsf3 binding may not be contributing to the splicing outcomes in these cases. Instead, we correlated the eCLIP peaks with AS events by identifying transcripts in which Srsf3 bound within an alternatively-spliced exon or within 250 bp of the neighboring introns. We have added additional text clarifying this point in the Results: “We next sought to identify high-confidence transcripts for which Srsf3 binding had an increased likelihood of contributing to AS. Previous studies revealed enrichment of functional RBP motifs near alternatively-spliced exons (Yee et al., 2019). As such, we correlated the eCLIP peaks with AS events across all four treatment comparisons by identifying transcripts in which Srsf3 bound within an alternatively-spliced exon or within 250 bp of the neighboring introns (Tables S12-S15).” Further, we have relabeled Figure 5B as “Highconfidence, overlapping datasets biological process GO terms”.

(5) In general, there is little validation of the sequencing results, performing qPCR on Arhgap12 and Cep55. The authors should additionally validate the PI3K pathway members that they analyze. Related, is Becn1 expression downregulated in the absence of Srsf3, as would be predicted if it is undergoing NMD?

We have added two new figure panels, Figure 5F-5G, assessing *Wdr81* AS and Wdr81 protein sizes, as this gene has previously been implicated in craniofacial development. We have added the following text to the Results section: “Finally, as Wdr81 protein levels are predicted to regulate RTK trafficking between early and late endosomes, we confirmed the differential AS of *Wdr81* transcripts between unstimulated scramble cells and scramble cells treated with PDGFAA ligand for 1 hour by qPCR using primers within constitutively-expressed exons flanking alternatively-spliced exon 9. This analysis revealed a decreased PSI for *Wdr81* in each of three biological replicates upon PDGF-AA ligand treatment (Fig. 5F). Relatedly, we assessed the ratio of larger isoforms of Wdr81 protein (containing the WD3 domain) to smaller isoforms (missing the WD3 domain) via western blotting. Consistent with our RNA-seq and qPCR results, PDGFAA stimulation for 24 hours in the presence of Srsf3 led to an increase in smaller Wdr81 protein isoforms (Fig. 5G).”

(6) What is the alternative splicing event for Acap3?

We have added the following text to the Results section and updated Figure 5E with *Acap3* eCLIP peak visualization and the predicted alternative splicing outcome: “Finally, Acap3 is a GTPase-activating protein (GAP) for the small GTPase Arf6, converting Arf6 to an inactive, GDP-bound state (Miura et al., 2016). Arf6 localizes to the plasma membrane and endosomes, and has been shown to regulate endocytic membrane trafficking by increasing PI(4,5)P2 levels at the cell periphery (D’Souza-Schorey and Chavrier, 2006). Further, constitutive activation of Arf6 leads to upregulation of the gene encoding the p85 regulatory subunit of PI3K and increased activity of both PI3K and AKT (Yoo et al., 2019)… Srsf3 binding was additionally increased in *Acap3* exon 19 upon PDGF-AA stimulation, at an enriched motif within the highconfidence, overlapping datasets, and we observed a corresponding increase in excision of adjacent intron 19 (Fig. 5D,E). As *Acap3* intron 19 contains a PTC, this event is predicted to result in more transcripts encoding full-length protein (Fig. 5E).”

(7) The insets in Figure 6 C"-H" are useful but difficult to see due to their small size. Perhaps these could be made as their own figure panels.

We have increased the size of the previous insets in new Figure 6 panels C’’’-H’’’.

(8) In Figure 6A, it is not clear which groups have statistically significant differences. A clearer visualization system should be used.

We have added bracket shapes to Figure 6A indicating the statistically significant differences between scramble 0 minutes and scramble 60 minutes, and between scramble 60 minutes and shSrsf3 60 minutes.

(9) Similarly in Figure 6B, is 15 vs 60 minutes in the shSrsf3 group the only significant difference? Is there a difference between scramble and shSrsf3 at 15 minutes? Is there a difference between 0 and 15 minutes for either group?

We have added a bracket shape to Figure 6B indicating the statistically significant difference between shSrsf3 at 15 minutes and shSrsf3 at 60 minutes. No other pairwise comparisons between treatments or timepoints were statistically significantly different.

**Reviewer #2 (Public Review):**
Summary:This manuscript builds upon the work of a previous study published by the group (Dennison, 2021) to further elucidate the coregulatory axis of Srsf3 and PDGFRa on craniofacial development. The authors in this study investigated the molecular mechanisms by which PDGFRa signaling activates the RNA-binding protein Srsf3 to regulate alternative splicing (AS) and gene expression (GE) necessary for craniofacial development. PDGFRa signaling-mediated Srsf3 phosphorylation drives its translocation into the nucleus and affects binding affinity to different proteins and RNA, but the exact molecular mechanisms were not known. The authors performed RNA sequencing on immortalized mouse embryonic mesenchyme (MEPM) cells treated with shRNA targeting 3' UTR of Srsf3 or scramble shRNA (to probe AS and DE events that are Srsf3 dependent) and with and without PDGF-AA ligand treatment (to probe AS and DE events that are PDGFRa signaling dependent). They found that PDGFRa signaling has more effect on AS than on DE. A matching eCLIP-seq experiment was performed to investigate how Srsf3 binding sites change with and without PDGFRa signaling.Strengths:(1) The work builds well upon the previous data and the authors employ a variety of appropriate techniques to answer their research questions.(2) The authors show that Srsf3 binding pattern within the transcript as well as binding motifs change significantly upon PDGFRa signaling, providing a mechanistic explanation for the significant changes in AS.(3) By combining RNA-seq and eCLIP datasets together, the authors identified a list of genes that are directly bound by Srsf3 and undergo changes in GE and/or AS. Two examples are Becn1 and Wdr81, which are involved in early endosomal trafficking. We thank the Reviewer for these supportive comments.Weaknesses:(1) The authors identify two genes whose AS are directly regulated by Srsf3 and involved in endosomal trafficking; however, they do not validate the differential AS results and whether changes in these genes can affect endosomal trafficking. In Figure 6, they show that PDGFRa signaling is involved in endosome size and Rab5 colocalization, but do not show how Srsf3 and the two genes are involved.

We have added two new figure panels, Figure 5F-5G, assessing *Wdr81* AS and Wdr81 protein sizes, as this gene has previously been implicated in craniofacial development. We have added the following text to the Results section: “Finally, as Wdr81 protein levels are predicted to regulate RTK trafficking between early and late endosomes, we confirmed the differential AS of *Wdr81* transcripts between unstimulated scramble cells and scramble cells treated with PDGFAA ligand for 1 hour by qPCR using primers within constitutively-expressed exons flanking alternatively-spliced exon 9. This analysis revealed a decreased PSI for *Wdr81* in each of three biological replicates upon PDGF-AA ligand treatment (Fig. 5F). Relatedly, we assessed the ratio of larger isoforms of Wdr81 protein (containing the WD3 domain) to smaller isoforms (missing the WD3 domain) via western blotting. Consistent with our RNA-seq and qPCR results, PDGFAA stimulation for 24 hours in the presence of Srsf3 led to an increase in smaller Wdr81 protein isoforms (Fig. 5G).” The experiments in Figure 6 compare early endosome size, PDGFRa localization in early endosomes and phospho-Akt levels in response to PDGF-AA stimulation in scramble versus shSrsf3 cells, demonstrating that Srsf3-mediated PDGFRa signaling leads to enlarged early endosomes, retention of PDGFRa in early endosomes and increased downstream phospho-Akt signaling. Though we agree with the Reviewer that functionally linking the AS events to the endosomal phenotype would strengthen our conclusions, these are technically challenging experiments for several reasons. First, this approach has typically relied on tiling oligos against a region of interest to find the optimal sequence. We identified several transcripts that are bound by Srsf3 and undergo alternative splicing upon PDGFRa signaling to potentially contribute to the regulation of PI3K signaling and early endosomal trafficking. We do not expect that these effects are mediated by a single transcript but may instead by mediated by a combination of alternative splicing changes. As such, these experiments would require us to identify and validate multiple splice-switching antisense oligonucleotides (ASOs). Second, ASOs designed against a specific target may not lead to alternative splicing of that target, even in cases of high predicted binding affinities (Scharner et al., 2020, *Nucleic Acid Res* 48(2), 802816). Third, ASOs have been shown to result in off-target mis-splicing effects, which are hard to predict (Scharner et al., 2020, *Nucleic Acid Res* 48(2), 802-816). The design of functional ASOs is thus a long-standing challenge in the field, and likely beyond the scope of this manuscript. We have added the following text to the Discussion to highlight this potential future direction: “In the future, it will be worthwhile to attempt to functionally link the AS of transcripts such as *Becn1, Wdr81* and/or *Acap3* to the endosomal trafficking changes observed above using spliceswitching antisense oligonucleotides (ASOs).”

(2) The proposed model does not account for other proteins mediating the activation of Srsf3 after Akt phosphorylation. How do we know this is a direct effect (and not a secondary or tertiary effect)?

This point is introduced in the Discussion: “Whether phosphorylation of Srsf3 directly influences its binding to target RNAs or acts to modulate Srsf3 protein-protein interactions which then contribute to differential RNA binding remains to be determined, though findings from Schmok et al., 2024 may argue for the latter mechanism. Studies identifying proteins that differentially interact with Srsf3 in response to PDGF-AA ligand stimulation are ongoing and will shed light on these mechanisms…. Again, this shift could be due to loss of RNA binding owing to electrostatic repulsion and/or changes in ribonucleoprotein composition and will be the subject of future studies.” We have added a potential change in Srsf3 protein-protein interactions upon Akt phosphorylation in the model in Figure 6J.

**Reviewer #2 (Recommendations For The Authors):**
Suggestions:(1) It would strengthen the paper and improve the connection with the other sections of the paper if the authors show:a) validation of PDGFRa signaling leading to AS of Becn1 and Wdr81 and corresponding changes in protein, and

We have added two new figure panels, Figure 5F-5G, assessing *Wdr81* AS and Wdr81 protein sizes, as this gene has previously been implicated in craniofacial development. We have added the following text to the Results section: “Finally, as Wdr81 protein levels are predicted to regulate RTK trafficking between early and late endosomes, we confirmed the differential AS of *Wdr81* transcripts between unstimulated scramble cells and scramble cells treated with PDGFAA ligand for 1 hour by qPCR using primers within constitutively-expressed exons flanking alternatively-spliced exon 9. This analysis revealed a decreased PSI for *Wdr81* in each of three biological replicates upon PDGF-AA ligand treatment (Fig. 5F). Relatedly, we assessed the ratio of larger isoforms of Wdr81 protein (containing the WD3 domain) to smaller isoforms (missing the WD3 domain) via western blotting. Consistent with our RNA-seq and qPCR results, PDGFAA stimulation for 24 hours in the presence of Srsf3 led to an increase in smaller Wdr81 protein isoforms (Fig. 5G).”

b) functionally link the AS event(s) to endosomal phenotype using ASOs, etc.

Though we agree with the Reviewer that such results would strengthen our conclusions, these are technically challenging experiments for several reasons. First, this approach has typically relied on tiling oligos against a region of interest to find the optimal sequence. We identified several transcripts that are bound by Srsf3 and undergo alternative splicing upon PDGFRa signaling to potentially contribute to the regulation of PI3K signaling and early endosomal trafficking. We do not expect that these effects are mediated by a single transcript but may instead by mediated by a combination of alternative splicing changes. As such, these experiments would require us to identify and validate multiple splice-switching antisense oligonucleotides (ASOs). Second, ASOs designed against a specific target may not lead to alternative splicing of that target, even in cases of high predicted binding affinities (Scharner et al., 2020, *Nucleic Acid Res* 48(2), 802-816). Third, ASOs have been shown to result in off-target mis-splicing effects, which are hard to predict (Scharner et al., 2020, *Nucleic Acid Res* 48(2), 802-816). The design of functional ASOs is thus a long-standing challenge in the field, and likely beyond the scope of this manuscript. We have added the following text to the Discussion to highlight this potential future direction: “In the future, it will be worthwhile to attempt to functionally link the AS of transcripts such as *Becn1, Wdr81* and/or *Acap3* to the endosomal trafficking changes observed above using splice-switching antisense oligonucleotides (ASOs).”

(2) The Venn diagram in Figure 5A and the description of the analysis the authors did to combine the RNA-seq and eCLIP-seq data are a little confusing. The authors say that they correlated eCLIP peaks with GE or AS events across all four treatment comparisons. The purpose of looking at both datasets was to find genes that are directly bound by Srsf3 and also have significantly affected GE and/or AS. Therefore, the data with and without PDGF-AA should be considered separately. For example, eCLIP peaks in the PDGF-AA condition can be correlated to Srsf3-dependent AS differences (comparing shSrsf3 and scramble) in the -PDGF-AA condition, and eCLIP peaks in the +PDGF-AA condition can be correlated to Srsf3-dependent AS differences in the +PDGF-AA condition. In the Venn diagram and the description, it seems like all comparisons were combined and it is not clear how the data were analyzed.

As indicated in Figure 5A, 233 of the alternatively-spliced transcripts uniquely found in one of the four treatment comparisons had an Srsf3 eCLIP peak. However, several of these eCLIP peaks were a large distance from an alternatively-spliced element in the rMATS datasets, indicating that Srsf3 binding may not be contributing to the splicing outcomes in these cases. Instead, we correlated the eCLIP peaks with AS events by identifying transcripts in which Srsf3 bound within an alternatively-spliced exon or within 250 bp of the neighboring introns. We have added additional text clarifying this point in the Results: “We next sought to identify highconfidence transcripts for which Srsf3 binding had an increased likelihood of contributing to AS. Previous studies revealed enrichment of functional RBP motifs near alternatively-spliced exons (Yee et al., 2019). As such, we correlated the eCLIP peaks with AS events across all four treatment comparisons by identifying transcripts in which Srsf3 bound within an alternativelyspliced exon or within 250 bp of the neighboring introns (Tables S12-S15).” Further, we have relabeled Figure 5B as “High-confidence, overlapping datasets biological process GO terms”. We respectfully disagree with the Reviewer’s suggested comparisons. A comparison of the PDGF-AA eCLIP data with the scramble vs shSrsf3 (-PDGF-AA) data from the list of highconfidence transcripts resulted in only 7 transcripts. Similarly, a comparison of the +PDGF-AA eCLIP data with the scramble vs shSrsf3 (+PDGF-AA) data from the list of high-confidence transcripts resulted in only 14 transcripts. Separate gene ontology analyses of these lists of 7 and 14 transcripts revealed 21 and 40 significant terms for biological process, respectively, the majority of which encompassed one, and never more than two, transcripts. Had we separately examined the -PDGF-AA and +PDGF-AA data, we would not have detected the changes in *Becn1, Wdr81* and *Acap3* in Figure 5E.